# ETS transcription factor pointed controls germline survival in *Drosophila*

**Alicia E. Rosales-Nieves**[¤a☺‡*], **Miriam Marín-Menguiano**[☺], **Lourdes López-Onieva**[¤b], **Juan Garrido-Maraver**, **Acaimo González-Reyes**[*]

Centro Andaluz de Biología del Desarrollo, CSIC/Universidad Pablo de Olavide/JA, Sevilla, Spain

☺ These authors contributed equally.
‡ Lead contact.
¤a Current address: Instituto de Biomedicina de Sevilla (IBiS) Hospital Universitario Virgen del Rocío/CSIC/Universidad de Sevilla, Departamento de Bioquímica y Biología Molecular, Facultad de Farmacia, Sevilla, Spain
¤b Current address: Departamento de Bioquímica y Biología Molecular I, Facultad de Ciencias, Universidad de Granada, Centre for Genomics and Oncological Research (GENYO), Instituto de Investigación Biosanitaria ibs.GRANADA, Hospital Virgen de las Nieves, Granada, Spain
* aerosales-ibis@us.es (AERN); agonrey@upo.es (AGR)

## Abstract

Proper gonad development is a pre-requisite for gametogenesis and reproduction. During female gonad formation in *Drosophila*, the EGF receptor (EGFR) signalling pathway ensures the correct number of primordial germ cells (PGCs) populate the larval gonad. We study the gene *pointed (pnt)*, which acts downstream of the EGFR receptor and belongs to the ETS transcription factor family, with a previously unknown function in gonadogenesis. We report that *pnt* is expressed in female larval gonads and later in the adult ovarian germline niche and that it is required to sustain proper gametogenesis. Loss of *pnt* function in female larval gonads, similar to the EGFR, induced PGC overproliferation. Conversely, we isolated a novel mutant allele gene, termed *pnt^aga^*, which resulted in agametic gonads and ovaries. While *pnt^aga^* embryos developed gonads containing a normal complement of PGCs, these are subsequently lost by apoptosis during late larval and pupal stages. Molecular characterization of *pnt^aga^* revealed reduced expression levels of the different *pnt* isoforms, unveiling a complex autoregulatory network involving the three Pnt proteins. We propose that germline survival in *Drosophila* gonads requires a precise tuning of EGFR signalling to ensure the appropriate transcriptional activation of its target *pnt*.

## Author summary

The germline is the group of cells in complex organisms that enables sexual reproduction. Before forming the adult reproductive organs, populations of these cells transition from an undifferentiated state—known as primordial germ

provided the original author and source are credited.

**Data availability statement:** All raw reads and summary files are available in GEO under the accession numbers: GSE301339, GSM9081228, GSM9081229, GSM9081230, GSM9081231, GSM9081232, GSM9081233.

**Funding:** This work was supported by the Spanish Agencia Estatal de Investigación (MCUI/AEI, http://www.ciencia.gob.es/; grant numbers PID2021-125480 NB-I00, MDM-2016-0687 and CEX2020-001088-M to AG-R), by the Junta de Andalucía (grant number P20_00888 to AG-R) and by the European Regional Development Fund (http://ec.europa.eu/regional_policy/en/funding/erdf/). Core funding to the CABD from the Junta de Andalucía is acknowledged. The funders had no role in study design, data collection and analysis, decision to publish, or preparation of the manuscript.

**Competing interests:** The authors have declared that no competing interests exist.

cell—to a committed state in which they behave as germline stem cells, initiating germline differentiation and eventually producing functional gametes. In this study, we used the fruit fly *Drosophila melanogaster* to investigate early ovary development. We identified a novel function for the *Drosophila* gene *pointed* in the female larval gonad. Making use of a new mutation in the locus, which we describe in detail in the text and which causes germline loss in both ovaries and testes, we report a regulatory feed-back loop between the three known Pointed isoforms. This work provides the first evidence that *pointed* plays an essential role in germline proliferation, a finding of particular significance because *pointed* belongs to a conserved family of ETS transcription factors with counterparts in humans.

## 1.  Introduction

Understanding how tissue specification and morphogenesis are controlled is a central question in biology. During animal development, multiple processes are tightly orchestrated in time and space to facilitate specific tissue differentiation and organ formation. *Drosophila* oogenesis serves as an elegant model system in this regard, allowing for extensive analysis of cell behavior and organ morphogenesis in a genetically tractable and experimentally flexible context. Here, we report the isolation of a new mutation in the *pointed* (*pnt*) gene that halts germline development and provides key insights into morphogenesis.

Pointed belongs to the ETS-domain family, one of the largest groups of transcription factors described so far [1]. This family of proteins, unique to metazoans, encompasses over 30 proteins from humans to *Drosophila*. ETS (E26 Transformation Specific) transcription factors are implicated in critical developmental processes as well as in cancer progression, and they have been known to regulate a myriad of cellular and viral genes [1,2]. They are characterized by the presence of an evolutionary conserved DNA-binding domain (the ETS domain) and act in cooperation with a variety of structurally unrelated transcription factors and co-factors [3]. Most ETS proteins are nuclear targets of Ras-MAP kinase signaling cascades and regulate cell proliferation, differentiation and apoptosis by activating genes encoding growth factor receptors and integrin families, among others [4,5]. Furthermore, studies in model organisms such as *C. elegans* and *Drosophila* have uncovered the critical role of these transcription factors in a variety of developmental processes such as oogenesis, metamorphosis, neurogenesis, eye development and myogenesis [6].

The *Drosophila pointed* locus gives rise to four transcription variants: *pnt-RB, -RC, -RD* and *-RE*. Since *pnt-RC* and *pnt-RE* only differ in their 3'UTR, the locus gives rise to three different protein isoforms: Pnt-PB, Pnt-PD and Pnt-PC/E (hereafter we will refer to the three products as long, intermediate and short; Fig 1A and 1B). The three Pnt proteins share an ETS domain, common to all members of the ETS family [7]. However, only Pnt$^{long}$ and Pnt$^{intermediate}$ incorporate a second

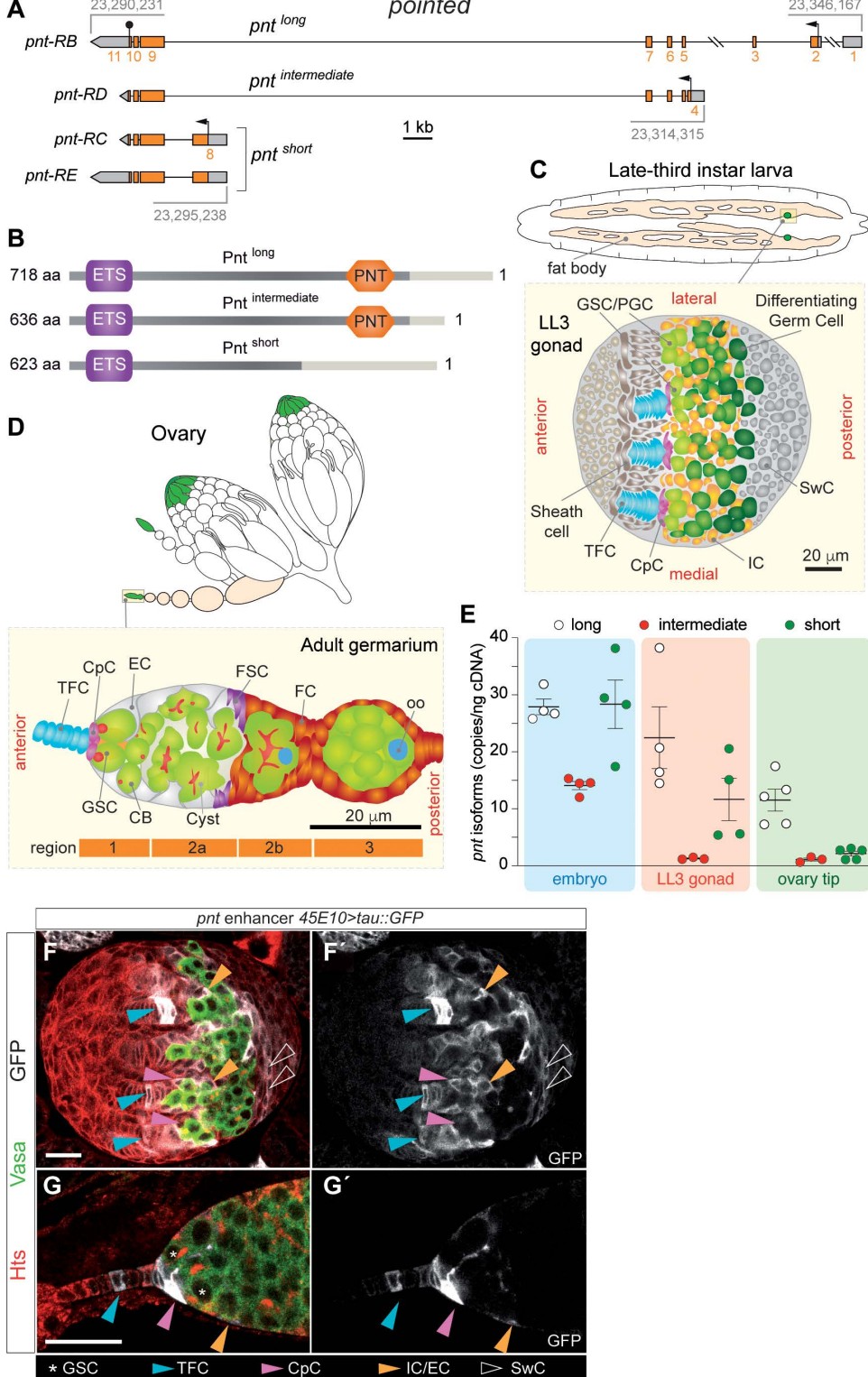

**Fig 1. _pointed_ is expressed in female larval gonads and in the ovarian GSC niche. (A)** Scheme of the _pointed_ locus highlighting the four predicted isoforms. Arrows designate transcriptional start sites. The stop codon is indicated by the black sphere. UTRs (untranslated regions) are shown in grey. The three different ORFs (open reading frames) are coloured in orange and the exons are numbered in orange. Numbers in grey refer to genomic

coordinates according to FlyBase (release FB2024_02). **(B)** Scheme of the three Pointed protein isoforms. The PNT (Pointed) and the ETS (E26-Transformation Specific) domains are highlighted. Light grey boxes represent isoform-specific domains. **(C)** Drawing of a female LL3 larva showing the fat body and the medial position of the embedded gonads. Drawing of an LL3 gonad revealing the distribution of relevant cell types. **(D)** Drawing of a pair of ovaries to highlight the organization of ovarioles and the localization of germaria at their tip. Drawing of a germarium to show the distribution of relevant cell types. **(E)** Detection of the *pnt-RB* (long), *-RD* (intermediate) and *-RC/E* (short) mRNAs using droplet-digital PCR in control embryos, LL3 gonads and ovary tips containing germaria and few early egg chambers. The primer pairs specific for each of the three isoforms map to exons 2 and 5 (long), 4 and 5 (intermediate) and 8 and 9 (short). Measurements correspond to at least three biological replicates and to one or two technical replicates. **(F, G)** Pattern of expression of the *pointed* 45E10 enhancer in LL3 gonads (F) and in germaria (G). 45E10-driven expression of the Tau::GFP reporter is shown in white, Hts in red and Vasa in green. Scale bars: 20 μm. In this and the rest of the figures, abbreviations used are the following: TFC: terminal filament cell; CpC: cap cell; EC: escort cell; GSC: germline stem cell; PGC: primordial germ cell; CB: cystoblast; FSC: follicle stem cell; FC: follicle cell; oo: oocyte; IC: intermingled somatic cell; SwC: Swarm cell. Related to S1 Fig.

conserved homology domain named POINTED (PNT), also known as SAM-PNT (for Sterile Alpha Motif), which becomes phosphorylated upon activation of the upstream Ras-MAPK pathway (Fig 1B) [8–10]. Further allelic characterization of *pnt^long* promoter described its function during eye and wing development and identified two sequence domains with opposite effect on transcription, one *trans*-activating and another *trans*-repressing transcription of *pnt^long* [11]. Contrariwise, Pnt^short is constitutively active because it lacks the phosphorylation domain. The original model suggested that Pnt^long, which responds to the EGFR/Ras/MAPK pathway, may turn on the *pnt^short* promoter, thus allowing a prolonged activation signal in the target cell [9]. Both isoforms typically work in concert with their transcriptional partner, Yan, another ETS-related transcription factor encoded by the *yan* gene, which serves as a negative regulator [9,12–14]. Specifically in the context of eye development, Yan operates in opposition to the Ras signalling function [9,15]. It was originally proposed that Yan competed with Pnt for binding targets [9]. More recently, however, it has been suggested that Pnt promotes the recruitment and stabilization of Yan and co-repressor Groucho. Upon activation of EGFR signalling, disassembly of the Yan, Pnt and Gro complex enables RNA polII transcription elongation [13]. The highly conserved ETS domain binds DNA over a range of 12–15 bp, exhibiting sequence preference for around 9 bp with a central invariable core: 5'-GGA(A/T)-3' [16]. The transcriptional activity of the *Drosophila* Pnt^long, and its mammalian orthologs ETS1 and ETS2, depends on mitogen-activated protein (MAP) kinase phosphorylation. The PNT domain works as a docking sequence to enhance phosphorylation of an immediately adjacent threonine by, in the case of *Drosophila* Pnt^long, the MAP kinase Rolled [15].

In *Drosophila* females, the Germline Stem Cell (GSC) niche begins to shape at larval stages when the gonad consists of a growing, round mass of somatic connective tissue and primordial germ cells (PGCs), surrounded by fat body located posteriorly (Fig 1C). Third larval instar gonads house PGCs and precursors of several somatic cell types including the terminal filament cells (TFCs) and cap cells (CpCs), the interstitial cells — called intermingled cells (ICs) — and the swarm cells (SwCs; Fig 1C). The TFCs organize in stacks following a morphogenetic wave that moves medial to lateral and configures the final organization of the terminal filaments (TFs). The GSC niche is complete by early pupa, when TF formation is concluded and CpCs are incorporated at the base of the TFs [17–19]. During larval stages, PGCs proliferate, increasing their number from ~12 in the embryo to about 100 undifferentiated germline cells by mid-third instar larvae (ML3). PGCs duplicate numbers every 24h during first and second instar, then the rate decreases during the following 24h [20]. PGCs are prevented from differentiating until ML3, when enough PGCs exist to populate all the niches. The PGCs in contact with the CpCs cells at the base of the TFs will become Germline Stem Cells (GSCs), while the PGCs that end up away from the niche will enter differentiation. After ovariole individualization during pupal stages, the adult ovary is formed by 18–21 ovarioles where mature eggs are produced. At the anterior end of each ovariole, a conical structure called germarium harbors the GSC niche, a tightly regulated system of 2–4 stem cells and 3 different somatic cell types - CpCs, in direct contact with the GSCs; TFCs, at the very tip, forming the TF; and anterior escort cells (ECs) (Fig 1D). The GSC niche provides the signaling required for GSC maintenance and proliferation, allowing the generation of mature eggs throughout the lifetime of the fly.

The first characterizations of the *pnt* locus in *Drosophila* reported that it is required for proper central nervous system development (CNS) in embryos and for the morphogenesis of compound eyes. Embryonic *pnt* loss-of-function phenotypes include fusion of the CNS commissures and abnormal glial-neuron interactions [8,11]. In the adult, non-lethal allelic combinations of *pnt* present a rough eye phenotype, which could be indicative of defects in cell fate acquisition [11]. In fact, Pnt is a target of Ras signaling controlling photoreceptor determination and it is activated *in vitro* by MAP kinase phosphorylation [9,15]. In this study, we investigate the role of *pnt* in the context of *Drosophila* gametogenesis. We characterize a new mutation in the *pointed* locus that is homozygous viable but causes sterility, as the reproductive system of mutant adults is agametic. Our results demonstrate the importance of the correct transcription of the locus for proper gonadogenesis and hence gametogenesis and reproduction.

## 2. Results

### Pointed is expressed in the somatic cells of the female larval gonad and ovarian GSC niche.

*pnt* spans ≈56 kb and encodes four different transcriptional variants. The transcription start sites (TSS) of *pnt^long^* and *pnt^short^*, are separated by ≈50 kb of mostly intronic sequence. The TSS of *pnt^intermediate^* falls within this interval (Fig 1A). Considering the essential function of the EGFR pathway for proper female gonadogenesis and that *pnt* is a downstream target of the pathway [10,20], we studied a potential role for *pointed* in regulating GSC proliferation and/or niche formation throughout development. We used droplet digital PCR (ddPCR) to measure expression levels of the long, intermediate and short isoforms in embryos, female larval gonads and during early oogenesis in the adult. All three isoforms were found in embryos, consistent with the known roles of *pnt* in embryonic central nervous system development [8,21]. In contrast, female late third instar larval (LL3; 5 days after egg laying) gonads only expressed *pnt* short and long, while in ovary tips (germaria plus few, early egg chambers) we could only detect consistent levels of *pnt* long (Fig 1E).

To analyze further *pointed* expression in LL3 female gonads and throughout adult oogenesis, we utilized a number of lines expressing the Gal4 trans-activator under the control of different *pnt* enhancers and monitored their expression using a *UASp-Tau::GFP* reporter line to distinguish clearly cell shapes. In all ten lines analyzed, we detected *pointed* expression in most populations of somatic cells of the larval gonad including subsets of TFCs (often those cells in contact with the CpCs), CpCs and SwC. In the adult ovary, *pnt-Gal4*-driven Tau::GFP was consistently observed in TFCs, especially in TFCs near the CpC rosette, CpCs and in anterior ECs in the germarium (Figs 1F, 1G and S1). In addition, and as described before [10,22,23]Click or tap here to enter text., we could also detect expression of *pnt* in follicle cells at the posterior of the egg chamber from stage 6, as shown for the *Gal4* line NP5370, inserted within the *pnt* locus (S1 Fig). Thus, *pnt* expression seems to be restricted to the somatic component of the female gonad and adult ovary.

### *pointed* restricts PGC numbers in the female gonad and ensures cyst encapsulation in the germarium.

*pnt* null alleles are embryonic lethal [11]. Thus, studying *pnt* function during gonad formation requires either genetic tools to manipulate gene function in groups of cells or the use of hypomorphic mutations that give rise to viable adults. To evaluate the role of *pnt* in early oogenesis, first we used the UAS/Gal4 system to induce RNAi-mediated gene silencing. We examined the effect of lowering *pnt* expression in the larval gonad and/or the adult germarium by expressing different RNAi lines (HMS01452, KK100473 and JF02227; S2A Fig) under the control of *c587-Gal4* (*c587 > pnt^RNAi^*), which is expressed in the somatic cells of the gonad [24] and germarium [25] (S2B, S2E and S2F Fig). We observed a significant increase in PGC numbers for any of the RNAi lines compared to controls in female larval gonads (Fig 2A–C; similar results were obtained for the *traffic jam-Gal4* line, also expressed in the somatic cells of the gonad; S2B and S2D Fig). Growing larvae at increasing temperatures (22, 25 and 29 °C) produced a gradation in the PGC overproliferation phenotype. Furthermore, scoring PGCs in mitosis with the phospho-histone H3 marker confirmed higher PGC proliferation in RNAi conditions (Fig 2C and 2D). These results support previous reports showing that decreasing EGFR signalling in the

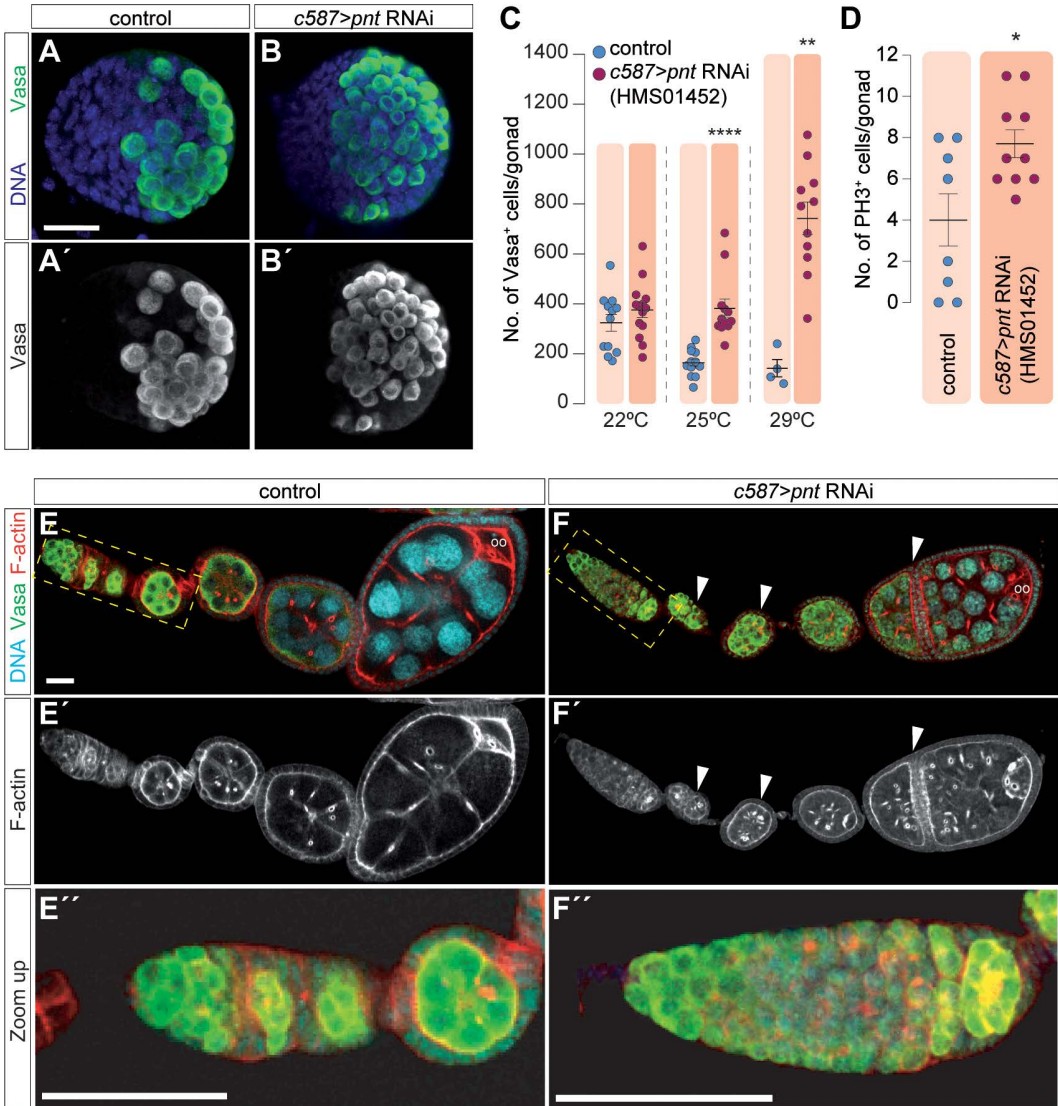

**Fig 2. *pointed* restricts PGC numbers in the gonad and ensures proper cyst encapsulation in the germarium. (A, A')** Control LL3 gonad stained with anti-Vasa (green) to show PGCs and with a DNA dye (blue) to mark chromatin. **(B, B')** LL3 gonad from a *c587>pnt RNAi^JF02227^* female larva showing the excess of PGCs found in *pnt* loss-of-function gonads. **(C)** Quantification of the number of Vasa+ cells in control and *c587>pnt RNAi^HMS01452^* at different temperatures in LL3 gonads. The arithmetic mean and the SEM are shown. **(D)** Quantification of the number of germ cells in mitosis using the PH3 marker. **(E, E')** Ovariole from a control female stained to visualize F-actin (red), the germline (anti-Vasa; green) and with a DNA dye (blue). Developing egg chambers are formed in the germarium and each contains a 16-cell germline cyst surrounded by a monolayered follicular epithelium. The oocyte is always found at the posterior of the cyst. **(F, F')** Ovariole from a *c587>pnt RNAi^JF02227^* female stained as in (D). White arrowheads point to fused egg chambers containing more than one oocyte and 15 nurse cells. **(E," F")** High magnifications of control and *c587>pnt RNAi^JF02227^* germaria. *** = p < 0.0005; **** = p < 0.0001. Scale bar: 20 μm. Related to S2 Fig.

somatic cells of the female larval gonad gave rise to excess PGCs [20,26]. Our findings thus suggest that *pnt* acts downstream of the EGFR pathway in the intermingled cells of the female gonad to control proliferation and survival of PGCs. Moreover, the expression of the JF02227 RNAi construct with the *c587-Gal4* driver in adult females gave rise to germaria larger than controls containing more germline cells, in agreement with a previous report [27]. In addition, we also observed egg-chamber fusions (Fig 2E and 2F), a novel phenotype suggesting that *pnt* is involved in somatic cell fate specification,

proliferation and/or migration in the germarium. Since the RNAi lines used in these experiments are designed to target regions common to all the *pnt* isoforms and since the *c587-* and *traffic jam-Gal4* lines are expressed in somatic cells (S2A, S2D and S2E Fig), we conclude from these results that reduction of *pnt* function in the somatic component of female gonads and ovaries controls germline proliferation and gametogenesis.

### *pnt*<sup>aga</sup>, a novel mutation that induces germline cell death.

We have isolated and characterized a spontaneous mutation in the *pnt* locus that we named *pnt agametic* (*pnt*<sup>aga</sup>). Contrary to the above results, adult *pnt*<sup>aga</sup> homozygous males and females were viable but displayed a highly penetrant agametic phenotype that rendered them sterile (Figs 3, S3A and S3B). This phenotype was not a consequence of abnormal embryonic gonad development, since gonads from control and *pnt*<sup>aga</sup> embryos were indistinguishable (S4 Fig). In contrast, the phenotypic characterization of ML3 gonads from female *pnt*<sup>aga</sup> larvae (4–5 days after egg laying) already showed significant differences with controls. Thus, compared to controls (average $98.22 \pm 5$ *n* (number of gonads analyzed) = 9), the mutant PGC pool had an aberrant and disorganized morphology and the number of mutant PGCs at ML3 was considerably reduced (average $55.5 \pm 4.35$ *n* = 10; Fig 3A, 3B and 3I). To analyze the consequences for gonad maturation of the effects observed in larval tissues, we stained female control and *pnt*<sup>aga</sup> white pupal gonads (~24 hours later in development) to find that mutant gonads contained greatly reduced PGC numbers compared to controls (controls: average $206.87 \pm 8.7$ *n* = 8; *pnt*<sup>aga</sup>: average $34 \pm 3.8$ *n* = 13). In addition, the few remaining mutant germ cells appeared aberrant in shape and larger in size (Fig 3C, 3D and 3I). Finally, homozygous mutant adult ovaries were considerably smaller than controls and were completely deprived of germ cells including GSCs, cystoblats and cysts. This phenotype was fully penetrant, as 100% of adult ovarioles were negative for the germline marker Vasa (Fig 3E-H and 3I). We also used an anti-Engrailed antibody to establish that mutant niches, while depleted of germline cells, seemed to form properly and to contain a normal pool of TFCs and CpCs (Fig 3G and 3H). This further confirmed that the reduced ovaries of agametic females maintain some characteristics of normal ovarian niches [28]

To test whether the agametic phenotype observed in mutant ovaries is due to PGC death concomitant with niche formation, we stained ML3 gonads with anti-Caspase 3 or DCP1, broadly used markers for the execution of apoptosis. We observed that, compared to controls, Casp3 and DCP1 were up-regulated in the middle region of the mutant gonad, particularly in PGC nuclei (Fig 3J, 3K, S3C and S3D). Altogether, our observations demonstrate that homozygous *pnt*<sup>aga</sup> germ cells die by apoptosis during larval and pupal development, resulting in the final, highly penetrant agametic phenotype. To rule out the possibility that apoptosis is also upregulated in other somatic cells, including the ICs and that PGCs die as a consequence of this, we studied in detail higher magnification views of mutant gonads and found that Caspase3 signal is generally restricted to PGCs (controls: $2.33 \pm 0.56$ (*n* = 6) apoptotic PGCs and $6.7 \pm 1.17$ (*n* = 6) apoptotic somatic cells; *pnt*<sup>aga</sup>: $15.13 \pm 1.76$ (*n* = 8) apoptotic PGCs and $5.75 \pm 1.5$ (*n* = 8) apoptotic somatic cells; Fig 3L). In addition, the use of the IC-specific marker Traffic-jam did not coincide with DCP1-positive cells. From the above results, we conclude that the initial steps of ovary morphogenesis take place correctly and that, as development progresses, *pnt*<sup>aga</sup> germ cells are lost from the mutant gonads. Hence, *pnt*<sup>aga</sup> uncovers a specific function for *pointed* in gonad development.

### *pnt*<sup>aga</sup> affects *pointed* transcription

Using deficiency mapping, genetic tests and genomic sequencing, we mapped the molecular lesion responsible for the *pnt*<sup>aga</sup> phenotype to a two-nucleotide change (AC to CT; FlyBase coordinates: 23,295,576–7) in an intronic region upstream of the TSS of *pnt*<sup>short</sup> (Figs 4A, S5 and S6). Knowing that the molecular lesion of *pnt*<sup>aga</sup> does not alter the coding sequence of any of the three Pnt proteins, it is likely that *pnt*<sup>aga</sup> causes PGC degeneration by altering transcript levels of the *pnt* locus in the somatic cells of the female gonad. To test this, we performed ddPCR to assess the relative mRNA levels of the different *pnt* isoforms in mutant LL3 larval gonads, and compared them to controls. We observed a significant reduction in the expression levels of both the long and short isoforms in *pnt*<sup>aga</sup> (controls: long isoform

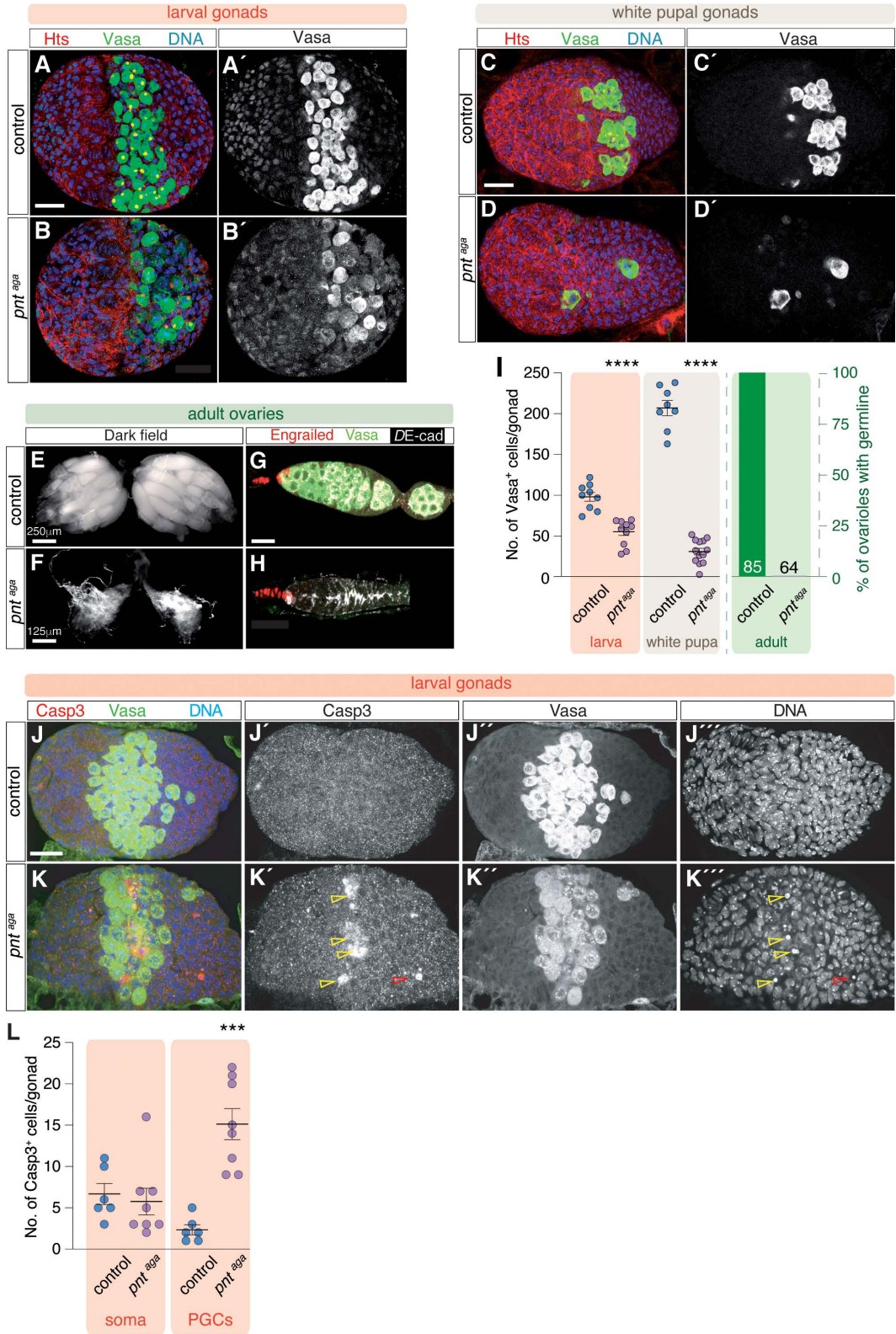

**Fig 3.** ***pnt***^aga^ **regulates germline survival. (A)** Control and **(B)** *pnt*^aga^ ML3 gonads stained with anti-Hts (red; to visualise cell outlines and germline spectrosomes and fusomes), anti-Vasa (green; to label the germline) and with a DNA dye (blue; to mark chromatin). **(C)** Control and **(D)** *pnt*^aga^ white pupa

gonads stained as in (A). Note the progressive decline in germline cells in the mutant condition. **(E, F)** Darkfield images of control and *pnt^aga* ovaries. **(G, H)** Control and *pnt^aga* germaria stained with anti-Engrailed (red; to label TFCs and CpCs), anti-Vasa (green) and *D*E-cad (white). Note that *pnt^aga* ovaries and germaria are devoid of germline cells. **(I)** Quantification of the number of Vasa+ cells in control and *pnt^aga* embryos, ML3 gonads and white pupa gonads. Quantification of the percentage of ovarioles with germline in control and mutant adult ovaries. Numbers in columns denote sample size (n). The arithmetic mean and the SEM are shown for each of the genotypes. **(J)** Control and **(K)** *pnt^aga* LL3 gonads stained with anti-Caspase 3 (red; to visualise apoptotic cells), anti-Vasa (green) and with a DNA dye (blue). **(L)** Quantification of the number of Casp3+cells in somatic or germline cells in control and *pnt^aga* gonads. \*\*\* = p < 0.0005; \*\*\*\* = p < 0.0001. Scale bars: 20 μm unless otherwise noted. Related to S3 and S4 Figs.

22.29 ± 10.8 copies/ng of cDNA; short 11.46 ± 7.5. *pnt^aga*: long 8.70 ± 0.6; short 3.60 ± 2.1. Two biological and two technical replicates per genotype; Fig 4B), demonstrating that *pnt^aga* represents a partial loss-of-function mutation. In fact, utilizing the expression of the diphosphorylated form of the extracellular signal-regulated kinase (dp-ERK) as a read-out of the EGFR pathway, we could confirm that the *pnt^aga* condition showed reduced dp-ERK levels (Fig 4C). Thus, *pnt^aga* impacts *pnt* transcription, leading to altered transcript levels of the different isoforms. Similarly, *pnt^aga* mutant embryos showed a decrease in long, intermediate and short isoforms (controls: long isoform 27.72 ± 2.7 copies/ng of cDNA; intermediate 13.89 ± 1.4; short 28.17 ± 8.5. *pnt^aga*: long 13.84 ± 3.7; intermediate 5.71 ± 0.8; short 13.47 ± 3.55. Two biological and two technical replicates per genotype; S4C Fig).

To determine whether an increase in *pnt* levels could rescue the *pnt^aga* phenotype, we generated flies containing *UAS-pnt* constructs in a *pnt^aga* background. We tested the three isoforms, long and short (*UASt-pnt^long*, *UASt-pnt^short*; [21]) and intermediate (*UASp-pnt^intermediate*; this work). In all instances, flies carrying any of the constructs were viable and fertile. Interestingly, *pnt^aga* females carrying two copies of any of the *UAS-pnt* constructs showed a significant proportion of ovaries containing germline (Fig 4D–G). Since *UAS* constructs can display some basal activity in absence of a *Gal4* driver [29], these phenotypic rescues confirmed the loss-of-function nature of the *pnt^aga* allele. Moreover, molecular analyses of *pnt* isoform levels upon combination with rescue constructs demonstrated a positive feedback loop regulating *pnt^long* transcription in experimental conditions over-expressing *pnt^short* or *pnt^intermediate* (Fig 4H). Next, we determined that *pnt* was required in somatic cells of the gonad, as the somatic *c587-Gal4* driver, but not the germline-specific *nanos-Gal4* (S2G and S2H Fig), was able to rescue the agametic phenotype when overexpressing *UASp-pnt^intermediate* (Fig 4G).

As stated above, the *pnt^aga* phenotype is opposite to that of *pnt^RNAi*, the former giving rise to gonads and germaria devoid of germ cells and the latter causing excess germ cell proliferation. Considering that *pnt^aga* LL3 gonads express reduced levels of long and short *pnt* and that *pnt^RNAi* knockdown reduces transcript levels of all isoforms (S2A Fig), it is possible that germline progression depends on the levels and/or combination of the different *pnt* isoforms expressed. In our working model, a strong reduction in *pnt* levels in somatic cells of the gonad (*c587 > pnt^RNAi*) impairs EGFR signaling and thus results in germ cell overproliferation, while a novel combination of long and short *pnt* induces germline disappearance by cell death (*pnt^aga*). If this hypothesis were right, the *pnt^aga* phenotype should be suppressed by the removal of all *pnt* function. To test this, we generated flies that carried the *pnt^RNAi* construct in a *pnt^aga* background (*c587 > pnt^RNAi + pnt^aga*) and scored the germline phenotype in adult females. We found that the expression of *pnt^RNAi* rescued partially the agametic phenotype typical of *pnt^aga*, as experimental *c587 > pnt^RNAi + pnt^aga* ovaries contained developing egg chambers in 38% of the cases (n = 34), whereas control *pnt^aga* showed complete absence of germline (n = 79; Fig 4I). From these results we interpret that germline development relies on the presence of the proper levels and combination of Pnt isoforms — mainly the long and short ones — in the somatic cells of the larval gonad. Thus, the RNAi-induced decrease in *pnt*-mediated signaling causes excess germ cell proliferation, most likely since this would mimic the abrogation of EGFR pathway activity in intermingled cells in the gonad [20]. Conversely, a new isoform combination of long and short *pnt,* as seen in the *pnt^aga* condition, induces the opposite effect, germline apoptosis.

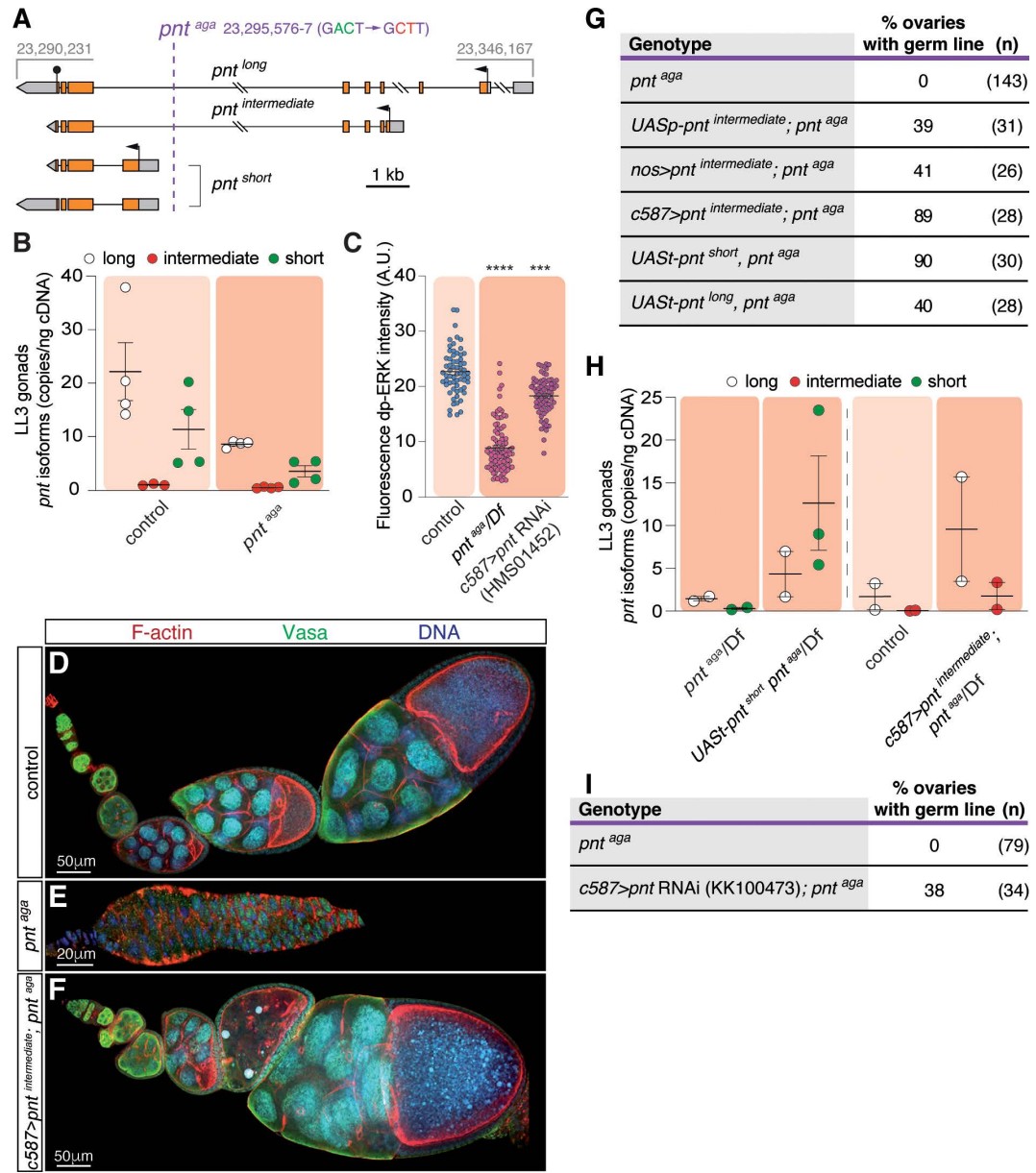

**Fig 4. pnt^aga affects pointed locus transcription. (A)** Mapping of the pnt^aga mutation. Numbers refer to genomic coordinates according to FlyBase (release FB2024_02). **(B)** Quantification of the long, intermediate and short pnt mRNA levels in control and pnt^aga gonads using droplet-digital PCR. Measurements correspond to three biological replicates and to one or two technical replicates. **(C)** Quantification of dp-ERK fluorescence intensity in control and c587>pnt RNAi^HMS01452 IC. **(D)** Control, **(E)** pnt^aga and **(F)** c587>pnt^intermediate; pnt^aga ovarioles stained to visualise F-actin (red), the germline (anti-Vasa; green) and with a DNA dye (blue). **(G)** Quantification of germline-containing ovaries in different genotypes. Ectopic expression in the soma of any of the pnt isoforms rescues the agametic phenotype. **(H)** Quantification of the long, intermediate and short pnt mRNA levels in control and pnt^aga/Df, UASt-pnt^short pnt^aga/Df, control and c587>pnt^intermediate; pnt^aga/Df gonads using droplet-digital PCR. Measurements correspond to two or three biological replicates and to one or two technical replicates. **(I)** Quantification of germline-containing ovaries in pnt^aga and in c587>pnt RNAi^KK100473; pnt^aga females. Reduction of pnt function in a pnt^aga background rescues partially the agametic phenotype. Scale bars: 20 or 50 μm, as indicated. Related to S4 and S5 Figs.

**A transcriptomic analysis of *pnt^aga* gonads identifies cell-type specific changes in cell adhesion, ion binding and membrane activity**

Considering the effect of *pnt^aga* on the various *pointed* isoforms, we aimed to determine how the altered levels of *pointed* transcripts influenced the global transcriptomics of mutant gonads. Using Affymetrix microarrays (GeneChip *Drosophila* Genome 2.0; > 13,500 genes arrayed), we performed a differential gene expression (DGE) analysis of control *vs* mutant LL3 gonads. We identified 253 genes whose expression was changed in *pnt^aga* gonads, 118 were upregulated and 135 down-regulated (Fig 5A and S1 Data; fold change<0.5 or >2; false discovery rate<5%). Gene Ontology (GO) enrichment analyses of the differentially expressed genes yielded a number of terms associated to cell adhesion or ion binding (up-regulated genes) or to membrane activity, including membrane transport, and response to mechanical stimulus (down-regulated genes), among others (Fig 5B).

The ETS family of transcriptions factors, comprising Pnt, are versatile in their functions and interact with other transcription factors and cofactors to allow combinatorial control of gene expression. As previously stated, the output of some Ras/MAPK signalling pathways depends upon the biochemical equilibrium between the partners Pnt and Yan, switching from an activation state of high Pnt to a contrasting inactivation state of high Yan. It is important to acknowledge that this bistable model depends on the specific cellular milieu and the intricacies of transcriptional regulation. Hence, a significant proportion of *pnt* targets most likely are tissue specific [30,31]. With this in mind, we compared the list of up- and down-regulated genes in *pnt^aga* gonads with known *pnt* targets in the eye disc (157 genes) and with *pnt*-bound genes (2584) in the embryo [13,31]. We found 8 genes in common with the eye disc targets and 44 also bound by Pnt in the embryo, with only 3 genes common to the three datasets (Fig 5C and S1 Data). While the identification of Pnt direct targets in the gonad requires further experimentation, from our results we conclude that the outcome of diminishing *pnt* transcript levels seems to be largely cell-type specific. Next, considering that the lower levels of long and short *pnt* observed in *pnt^aga* gonads indicated a partial loss of function situation, we compared the differentially expressed genes in *pnt^aga* with those identified in *pnt* null mutant embryos [13] and identified 7 common genes out of the ~9,000 genes spotted on the array [13]. Interestingly, 3 of the latter (*hibris, CG1773* and *3-Hydroxymethyl-3-methylglutaryl-CoA lyase*) were up-regulated in *pnt^aga*, while they were down-regulated in *pnt* null embryos, emphasizing the idea that *pnt*-mediated regulation of gene expression is context-specific. This view is further supported by the limited number (15) of direct *pnt* targets in the eye disc that were also differentially expressed in *pnt* null embryos and by the fact that only 1 gene was common to the three datasets (Fig 5D and S1 Data). Finally, we performed two statistical tests to calculate the probability of the overlap by chance between the different sets of genes, an analytical test (hypergeometric distribution) and an empirical test (permutation test with 10,000 permutations). These analyses dictated that the only significant overlap (p-value<0.05) in all of the pairwise comparisons was between the identified Pnt targets in the eye disc and the Pnt-bound genes in the embryo (p-value (analytical)=$9.99 \times 10^{-5}$; (empirical)=$4.83 \times 10^{-12}$) and between the Pnt eye targets and the differentially expressed genes in *pnt^null* embryos (p-value (analytical)=$1.16 \times 10^{-4}$; (empirical)=$3.00 \times 10^{-4}$). These results provide strong support for the context-dependent outcome of *pnt* transcriptional regulation.

## 3. Discussion

Soma-germline interactions are critical to control germline migration, proliferation, the induction and maintenance of germline stem cells, and gamete maturation. A number of pathways regulate gametogenesis, including the EGFR/Ras/MAPK, whose output depends on the activity of the Pnt and Yan cofactors. However, little is known about the role of *pnt* in gametogenesis. Our work, including the isolation and characterization of *pnt^aga*, demonstrates that *pointed* is required in female somatic gonadal cells to control germline proliferation and survival. For instance, a general reduction in *pnt* mRNAs levels utilizing RNA interference results in PGC overproliferation in gonads. On the contrary, the loss-of-function mutation *pnt^aga* produces gonads containing fewer PGCs than controls and this reduction in PGC number is due, at least

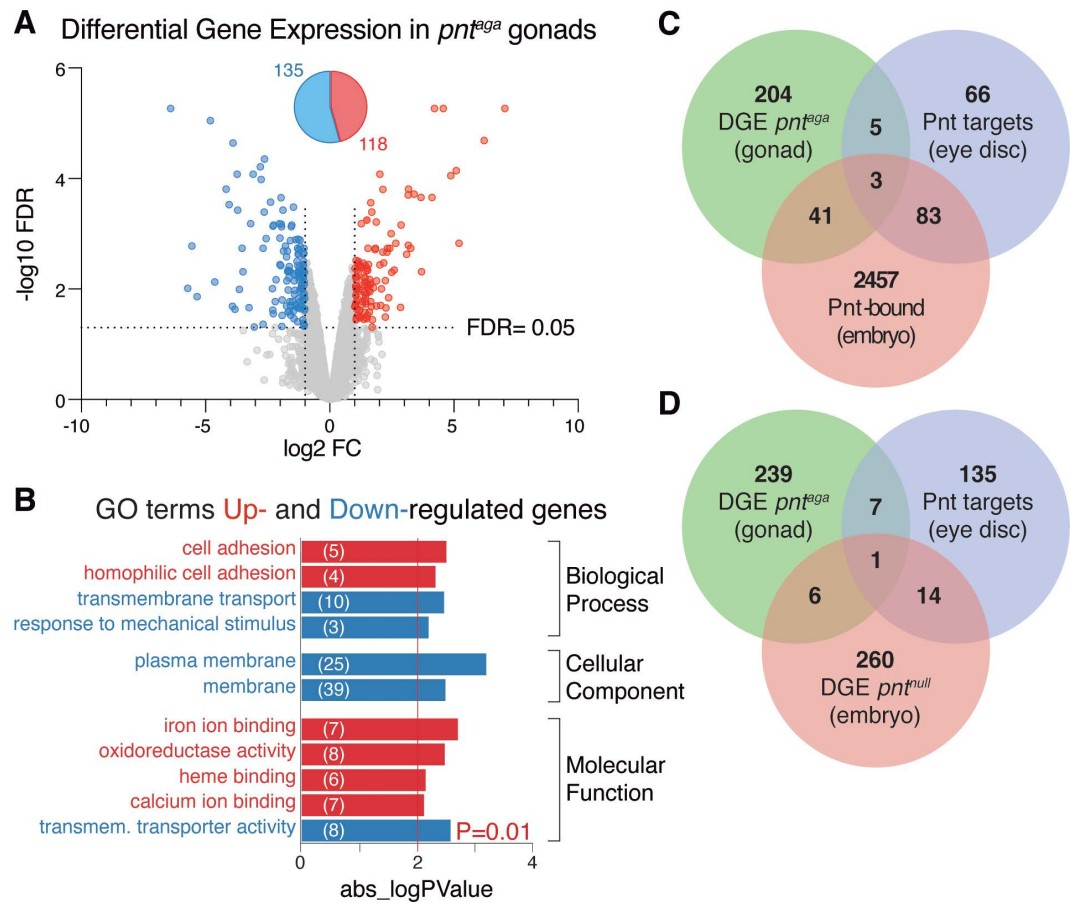

**Fig 5. Differential GE in *pnt^aga* gonads. (A)** Changes in gene expression (GE) — represented as log2 fold change of *pnt^aga*/control *versus* the -log10 false discovery rate (FDR) — identified 118 genes upregulated and 135 genes downregulated in *pnt^aga* compared to controls (linear fold change <0.5 or >2; FDR<5%). **(B)** GO terms identified (p-value<0.01) in the analysis of either up- or down-regulated genes. Numbers in parenthesis correspond to the genes identified in each term. **(C)** Diagram to indicate the number of genes showing differential GE in *pnt^aga* gonads common to the list of *pnt* targets in the eye disc [31] and the genes bound by Pnt in embryos [13]. **(D)** Diagram representing the number of differentially expressed genes in *pnt^aga* gonads and in *pnt^null* embryos, and of direct targets identified in the eye disc. Data related to *pnt^aga* correspond to three biological replicates per genotype. Related to S1 Data.

partially, to specific germ cell apoptosis. By the end of L3, mutant female gonads appear almost depleted of PGCs, thus preventing the establishment of germline stem cells. Hence, adult flies become agametic and oogenesis is disrupted. While at present we do not know the molecular mechanisms behind these opposite phenotypes, our work nevertheless highlights a critical function of *pnt* isoform combination in proper gonad development. *pnt^aga* maps to an intronic region just upstream of the TSS of the *pnt^short* isoforms. Based on its close proximity, we speculate that the molecular lesion in *pnt^aga* affects the promoter or an enhancer of *pnt^short* and thus its transcription. If this were the case, since *pnt^aga* also affects the transcription of *pnt^long* in female gonads and of *pnt^intermediate* and *pnt^long* in embryos, and since overexpression of either *pnt^short* or *pnt^intermediate* induces expression of *pnt^long* in gonads, our analyses suggest that feedback-loop mechanism(s) regulating *pnt* transcription under normal conditions are disrupted in *pnt^aga*. While we cannot rule out the possibility that *pnt^aga* affects a general enhancer for the locus and hence transcription from the three TSSs, because *pnt^aga* maps very close to *pnt^short* TSS and because *pnt^short* seemingly rescues the agametic phenotype more efficiently that *pnt^intermediate* or *pnt^long*, we favour the idea that *pnt^aga* affects specifically *pnt^short*.

It is known that the three Pnt proteins can perform different functions, as their distinct expression patterns, interactions and transcriptional activities are necessary for proper photoreceptor fate specification and thus eye patterning during larval stages [30]. These authors demonstrated that Pnt long and intermediate activate *pnt*short transcription. However, since they lacked a specific *pnt*short mutant, they could not assess regulatory interactions between Pnt short and *pnt*long and *pnt*intermediate. Our results in embryos and female gonads suggest that Pnt short might be needed to allow proper transcription from the long and intermediate promoters. Our evidence and that of other groups thus point to a complex autoregulatory network involving the three Pnt isoforms to ensure the correct transcriptional activity of the *pnt* locus [9,11,30]. This is particularly important in view of the critical function for *pnt* in germline development. Leveraging the drastically opposite phenotypes of a strong loss of function of the *pnt* locus (overproliferation of PGCs) and that of a ~50% reduction in *pnt*short and *pnt*long mRNA levels (agametic gonads), it follows that the correct amounts and/or combination of Pnt isoforms are critical to ensure the growth of functional female gonads. Moreover, while we cannot completely rule out the influence of other signalling pathways on *pnt* activity, the fact that *pnt RNAi* mimics EGFR loss-of-function phenotypes strongly suggests that precise regulation of the EGFR signalling pathway is essential for reproduction, at least in females. Although we have not investigated in detail the case of the male gonad and adult testes, the observation that *pnt*aga testes are agametic indicates that this regulatory mechanism may be conserved in males. A similar scenario in which different outcomes for the same signalling pathway depend on the dynamics of pathway activation has been reported in other tissues. For instance, JAK/STAT signaling in the male GSC niche shows contradictory outcomes, such as heterochromatin loss in somatic niche cells or, conversely, activation of heterochromatin in the GSCs [32,33]. In silico simulations predict that transient ligand stimulation, typical in somatic tissues, leads to a transitory decrease in unphosphorylated STAT levels, whereas continuous ligand stimulation, specific of the GSC niche, increases unphosphorylated STAT levels [34]. Another context in which the distinct expression of transcription factor (TF) isoforms can have profound consequences for tissue function is in cancer disease, where aberrant TFs' alternative splicing is linked to tumorigenesis. The altered composition of TFs may control different transcriptional programmes, regulate the same target genes with different efficiency or opposite effect, or compete with physiological TF activity [35,36].

The role of *pnt* in controlling embryonic and eye development in *Drosophila* has been known for decades [8,10,11,15,21]. More recently, novel direct *pnt* targets and the embryonic *pnt*/*yan* regulatory landscape have been reported [13,31]. *pnt*aga homozygous flies do not show obvious, visible phenotypes during development. Thus, other processes in which *pnt* plays a critical role appear to proceed normally. Considering that the agametic mutation also affects *pnt* transcription in embryos, but seemingly without compromising viability, we surmise that the gonad is particularly sensitive to *pnt* mRNAs levels of the long and short isoforms expressed there. Moreover, our results in the gonad also indicate that the cohort of genes responding to *pnt* activity is context-specific. In fact, there is little overlap between the genes that are regulated by *pnt* in the embryo, the eye disc or the *pnt*aga condition [13,31]. Our study thus identifies a critical function for *pnt* in gonadogenesis and contributes to the description of the *pnt* network in the developing gonad. Importantly, the essential role of *pnt* in gametogenesis seems conserved in mammals, as it parallels known functions of ETS-related proteins in the mammalian testes and ovaries [37–39]. In mammals, expression of the ETS related molecule (ERM) is restricted to somatic Sertoli cells and mice with a targeted mutation in ERM fail to maintain spermatogonial stem cells. Self-renewal of the niche seems to be disrupted during the first wave of spermatogenesis with a progressive depletion of the niche, producing a Sertoli-cell-only syndrome [39].

While single-cell techniques and bulk RNA sequencing have facilitated the transcriptional characterization of traditionally challenging tissues like the developing larval ovary [40], these omic approaches have their limitations. For instance, the recently published transcriptomic atlas of various cell types in the developing gonad did not report *pnt* expression in any somatic cells forming the niche or *egfr* transcription in ICs, despite the requirement of EGFR signalling in this cell type for PGC proliferation [20,40]. Therefore, direct genetic approaches serve as a valuable complement to broader strategies, aiding in the precise definition of the molecular functions of genes.

# 4. Material and methods

## Fly stocks and RNAi protocol

We used FlyBase (release FB2024_02) to find information on genes and their functions [41]. Flies were grown at 25°C on standard medium unless otherwise stated. The lines used include:

*TM3, twist-Gal4, UAS-nlsGFP* [42]

*UASp-tau::mGFP6* [43]

*pnt^{aga}* (this work)

*pnt^{Δ88}* [11] (Bloomington *Drosophila* Stock Centre (BDSC) #861)

P{w[+mC]=lacW} pnt^{1277} [11](BDSC #837)

P{w[+mW.hs]=GawB} pnt^{NP5370} (Kyoto Stock Centre #104978)

*tj-Gal4* [44]

*c587-Gal4* [45]

*nos-Gal4* [46] (BDSC #4442)

*UASp-pnt^{intermediate}* (this work)

*UASt-pnt.P1 (pnt^{short})* [21] (BDSC #869)

*UASt-pnt.P2 (pnt^{long})* [21] (BDSC #399)

FRT82B ubi-nls::GFP (BDSC #32655)

Df(3R)Exel6280, P{w[+mC]=XP-U}Exel6280 (BDSC #7746)

Df(3R)Exel9012, P{w[+mC]=XP-U}Exel9012 (BDSC #7990)

Df(3R)ED6105, P{w[+mW.Scer\FRT.hs3]=3'.RS5 + 3.3'}ED6105 (Kyoto Stock Center #150336)

P{TRiP.JF02227}attP2 (*UAS-pnt RNAi^{JF02227}*) (BDSC #31936)

*UAS-pnt RNAi^{KK100473}* (Vienna *Drosophila* Resource Center #105390)

y v;; P{TRiP.HMS01452}attP2 (*UAS-pnt RNAi^{HMS01452}*) (BDSC #35038)

The following enhancer-Gal4 constructs from the Janelia Farm collection were used (BDSC stock numbers in parenthesis): *P{GMR44C09-GAL4}attP2* (#41263), *P{GMR43E07-GAL4}attP2* (#45304), *P{GMR44B07-GAL4}attP2* (#45717), *P{GMR46C10-GAL4}attP2* (#46271), *P{GMR44C01-GAL4}attP2* (#48145), *P{GMR45D11-GAL4}attP2* (#49563), *P{GMR45F08-GAL4}attP2* (#49565), *P{GMR45B10-GAL4}attP2* (#50223), *P{GMR45E10-GAL4}attP2* (#50233).

## Generation of the UASp-pnt^{intermediate} construct

To express *pnt^{intermediate}* in the germline, we first synthesized the *pnt-RD* cDNA from 2-day old control (*y w*) ovaries. mRNA was isolated and purified using the QuickPrep Micro mRNA Purification Kit (GE healthcare) from approx. 100 ovary pairs. 1–2 µg of mRNA were used to synthesized cDNA with 0.5 µg of oligo(dT) (Sigma Genosys) and the Superscript II RNase H Transcriptase (Invitrogen LifeTechnology) in a 20 µl final volume. 2 µl of total cDNA were used in a PCR reaction to amplify the *pnt-RD* cDNA.

The following primers were used:

Forward primer

5' GCTAGGATCCATGACCAATGAGTGGATCGAT 3'

Reverse primer

5' GAATGCGGCCGCTTTGCGGTTGGTTGAGTACA 3'.

The *pnt-RD* cDNA generated by reverse transcription was then cloned as a BamHI/Not I fragment into pUASp [47]. The resulting construct was verified by restriction digests and sequencing. Transgenic lines were generated by standard procedures.

## Mapping *of* genomic deficiencies

Genomic DNA from single flies was prepared following the protocol by Georg Dietzl (Barry Dickson's Laboratory, IMP, Vienna). In short, one fly of the desired genotype was place in a 0.5 ml tube containing 5 µl of squishing buffer and mashed for 5–10 seconds with a pipette tip. After the addition of another 45 µl of squishing buffer, the mix was incubated at room temperature for 30 minutes. The Proteinase K was inactivated by heating the mix at 95°C for 1–2 minutes. Typically, 2 µl of supernatant were used per 20–50 µl PCR reaction.

Squishing buffer: 10 mM Tris-Cl pH 8.2, 1mM EDTA, 25 mM NaCl, 200 µg/ml Proteinase K (enzyme diluted fresh from a frozen stock).

The Exel6280 and Exel9012 deficiencies were generated by FLP/FRT deletions [48] and their breakpoints were originally mapped with reasonable precision and reported in FlyBase (flybase.org). They both harbour hybrid elements from the XP5' (plus and minus) and the WH5' P-elements, which we used to map the breakpoints in the *pnt* locus accurately. Internal primers corresponding to the XP elements were paired with genomic primers designed to bind to flanking sequences according to the published breakpoints. The PCR reaction would only amplify a DNA product from the chromosome bearing the deficiency, as internal primers are only complementary to the *P*-element sequences. To sequence the 3' genomic flanking region of *Df(3R)ED6105* we used a similar strategy, as this deficiency was generated by FLP/FRT-mediated recombination of two RS elements [49] In all cases, genomic DNA was prepared from Deficiency/Balancer flies, diluted 1:25 and used as template in PCR reactions. The primers used were the following (S6 Fig):

*Df(3R)Exel6280*, distal breakpoint:
XP3'+Flanking genomic region primers:
XP1 forward 5'GCCACCACTTCAAGAACTCTGTAGC 3'
G1 reverse 5'CGAGTTGGCGTTGTTAATGA 3'
*Df(3R) Exel9012*, distal breakpoint:
XP+Flaking genomic region:
52B forward 5' TTTACTCCAGTCACAGCTTTG 3'
G5 reverse 5' GCACAGTGGTTGTCATGGTC 3'
*Df(3R) ED6105*, distal breakpoint:
Pry4 (maps to the RS3r 3' terminal repeat end) forward 5' CAATCATATCGCTGTCTCACTCA 3'
Genomic reverse 5' GTTTGTGAGAGGCGGCAGCCA 3'

## Mapping of pnt<sup>aga</sup>

Genomic DNA from control *y w* and *pnt*<sup>aga</sup> homozygous flies was prepared as described, diluted 1:25 and used as template in PCR reactions. Genomic primers were designed based on the reference sequence in flybase.org. The following primers were used (S6 Fig):
Forward 5' AGAACGCGTTGTATGAGGGACCCA 3'
Reverse 5' GTTTGTGAGAGGCGGCAGCCA 3'

## Sample preparation, microarray hybridization and data analysis

RNA from 2 larval gonads (approximately 2000–3000 cells) per sample (Oregon-R controls and *pnt*<sup>aga</sup> homozygous; three biological replicas per genotype) was isolated using magnetic beads. cDNA synthesis, library preparation and amplification (Pico Profiling) are described elsewhere [50]. In short, after reverse-transcription each cDNA sample was added to an amplification mix that was subdivided into five equivalent parts for PCR amplification (26 cycles). Amplified cDNA collections were purified with the PureLink PCR Purification Kit (ThermoFisher Scientific), resuspended in 40µl and their concentrations measured using a Nanodrop 1000 spectrophotometer. The six cDNA collections were hybridized to GeneChip

*Drosophila* Genome 2.0 Arrays from Affymetrix. Raw data generated with Affymetrix's AGCC software were used as input for the DTT software. For the resulting datasets, RMA (Robust Microarray Average) normalization and linear modelling by *limma* (Linear Models for Microarray Analysis) were performed. The p-value for the False Discovery Rate (FDR) was ≤ 0.05 and fold change was set at <0.5 or >2. To test the overlap of different gene sets in Fig 5, we utilized the hypergeometric probability distribution and a permutation test with 10,000 permutations. We assumed a number of detectable *D. melanogaster* genes of 9,000, presuming that all of the ~9,000 genes present in the embryo microarray were also in the gonad microarray and that all were putative targets for the ChIP-seq analyses.

Homozygous *pnt*[aga] larvae were selected based on their loss of the Tubby marker on the TM6B balancer.

### Detection and quantification of mRNA levels by ddPCR

We used droplet digital PCR (ddPCR) to quantify mRNA levels of *pnt* long, intermediate and short isoforms from embryos, LL3 gonads and ovary tips (obtained after severing and collecting the germaria and few egg chambers from the whole ovaries). For embryos, RNA was isolated from ~30 embryos per genotype (*y w* controls and *pnt*[aga] homozygous; three biological replicas each; homozygous *pnt*[aga] embryos were selected based on their loss of the *TM3, twist-Gal4, UAS-GFP* chromosome) with RNAeasy Micro (Qiagen 74004) and QIAshredder (Qiagen 79654) columns. To synthesize cDNA, 10 µl of mRNA were incubated for 5 minutes at 65°C with 1 µl (0.5 µg) of Anchored Oligo(dT)$_{23}$ (Sigma O4387) + 1 µl of 10mM dNTP mix and then placed on ice for 1 minute. 4 µl of 5x First Strand buffer (Y02321) + 1 µl RNaseOUT Recombinant Ribonuclease Inhibitor (40 units/ µl) (100000840) + 2 µl 0.1 M DTT (Y00147) from Invitrogen were added and incubated for 1 minute at 42°C. Next, 1 µl of Super Script II RT from Invitrogen (18064–014) was added and incubated for 50 minutes at 42°C, 15 minutes at 70°C and then placed on ice for 1 minute. Finally, the mix was incubated for 20 minutes at 37°C with 1 µl of RNase H from Invitrogen (18021–014). For LL3 gonads, see above. Finally, to determine *pnt* expression in ovary tips, we dissected 30 adult ovaries per genotype and manually severed off the tips (which included the germarium and few young egg chambers per ovariole) using tungsten needles. RNA was isolated as in the case of embryos.

Primers used were the following (5'-3'): *pnt-RB* (long) isoform, forward (exon 2): TTCTGTCCAGCCTAGTTGAG, reverse (exon 5): AGTACCTCGTTGACCTTGCG; *pnt-RD* (intermediate) isoform, forward (exon 4): AACCACCAAT CATAGCAAGC, reverse (exon 5): AGTACCTCGTTGACCTTGCG; and *pnt-RC/E* (short) isoform, forward (exon 8): TGCGAATGCCTACTACACGG, reverse (exon 9): TACCGCTGCCATTGACAGTC. Ribosomal protein L32 (*RpL32*) and β-Tubulin (β-*Tub*) were used for normalization. *RpL32*, forward: ATGACCATCCGCCCAGCATAC, reverse: GCTTAG CATATCGATCCGACTGG; β-*Tub*, forward: GCAGTTCACCGCTATGTTCA, reverse: CGGACACCAGATCGTTCAT.

For the ddPCR, each reaction contained 10 µl of Master Mix ddPCR EVAGreen (Bio-Rad), 150 nM of each primer and 2,5 ng of cDNA template (except in the case of the housekeeping control, where we used 0,5 ng). Samples were prepared in duplicate with 10% additional volume. Droplet generation, PCR amplification and droplet analysis were done using the QX200 AutoDG ddPCR system (Bio-Rad). PCR conditions were: 95°C for 5 minutes (1x), 95°C for 1 minute and 65.2°C for 2 minutes (44x). For all steps a ramp rate of 2°C/s was used. Data were analysed with the Quantasoft software 1.7.4.0917 (Bio-Rad).

### Developmental staging of larvae and pupae

At 25°C, the end of embryogenesis is 24 hours after egg laying (AEL). The first and second larval instars are 24 hours each. Third larval instar lasts for 48 hours and puparium formation starts 120 hours AEL. In this study, ML3 (mid-third instar larvae) refers to larvae that crawled out of the food but that have not initiated pupariation. At this stage, most of TFs are still forming and cap cells are not ready visible. LL3 (late-third instar larvae) is used to identify larvae about to initiate pupariation. Their gonads present a number of already-formed TFs and some cap cells can be recognised. The

white pupa stage marks the initiation of pupariation and it is characterised by pale and clear pupae. At this stage, the GSC niches with their respective TFs and cap cell rosettes are well established [19].

## Immunohistochemistry

Adult flies were yeasted for 2 days before dissection in cold Ringer's. Ovaries and testes were then fixed for 10 minutes in 6% formaldehyde in fix buffer (600µl of heptane were added per 100µl of fix). After fixing, samples were permeabilized in 1% Triton in PBS for 2 hours and then blocked for 1 hour with PAT before overnight incubation with primary antibodies in PAT. The following morning primary antibodies were rinsed with PAT and the samples washed three times with PBT for 30 minutes. The incubation with secondary antibodies was done in PBT during 2–4 hours. Ovaries and testes were then washed three times with PTW, 10 minutes each. Mounting of testes and individual ovarioles was done in Vectashield medium (Vector Laboratories; Cat# H1000; RRID: AB_2336789).

To stain larval gonads, gonads embedded in the fat body were fixed in 5% formaldehyde in Ringer's for 20 minutes then washed for 5, 10 and 45 minutes with 1% PBT. The fat body was then blocked in 0.3% PBTB for 1 hour with gentle agitation. After blocking, the tissue was incubated overnight with the desired primary antibody diluted in 0.3% PBTB at 4℃ with gentle agitation. The next day the fat body was washed thrice (30 minutes each) with 0.3% PBTB. After blocking with 0.3% PBTB supplemented with 5% fetal bovine serum for 1 hour, the fat body was incubated for 2 hours with the secondary antibodies in blocking solution, followed by three washes of 30 min each with 0.3% PBT. Gonads were dissected from the fat body and mounted in Vectashield.

Embryos collected overnight on yeasted agar plates were dechorionated with bleach for 2 minutes and fixed for 20 minutes in 2 ml of 4% formaldehyde in PBS with an equal volume of heptane. After fixing, the aqueous phase was removed and 2 ml of methanol were added to remove the vitelline membrane. Devitellinized embryos were rinsed in methanol thrice and stored at -20℃. For antibody staining, embryos were re-hydrated in PBS:methanol for 5 minutes and then washed in PBS, in PBS:PAT and finally rinsed in PAT twice. Next, they were blocked for 1–2 hours in PAT at 4℃. Primary antibody incubation was done overnight at 4℃ in PAT, followed by a blocking step for several hours in PBT supplemented with 4% goat serum prior to secondary antibody incubation for 2–4 hours in PBT. Finally, embryos were washed thrice with PBT, 10 minutes each, and mounted in Vectashield.

Primary antibodies were used at the following concentrations: mouse anti-Hts (1B1, Developmental Studies Hybridoma Bank (DSHB) Cat# 1b1, RRID:AB_528070), 1:100; rabbit anti-Vasa 1:2000 (a gift from R. Lehmann); rat anti-DE-Cadherin (DSHB Cat# DCAD2, RRID:AB_528120) 1:100; mouse monoclonal anti-Engrailed (DSHB Cat# 4D9; RRID: AB_528224) 1:10; rat anti-Bab2 (a gift from F. Laski) 1:4000; rabbit anti-cleaved-Caspase3 (from Cell Signaling Technology) 1:50; goat anti-GFP FITC-conjugated (Abcam Cat# ab6662, RRID:AB_305635) 1:500; rabbit anti-dp-ERK (Cell Signaling) 1:20; Rabbit anti-Caspase DCP1 (Cell Signaling) 1:100. Secondary antibodies Alexa-Fluor 488, Cy2, Cy3 and Cy5 (Jackson Immuno Research Laboratories, Inc.; final concentrations of 1:100) and conjugated anti-GFP-488 nanobody (alpaca anti-GFP-Booster_Atto488 from ChromoTek, Cat# gba488–100, RRID:AB_2631386; final concentration 1:200) were incubated for four hours. To stain DNA, ovaries were incubated for 10 minutes with Hoechst (Sigma B2883 10 mg/ml in $H_2O$). To visualise F-actin, fixed ovaries were incubated in PBT + Rhodamine-labelled Phalloidin (Biotium Cat# BT-00027; final concentration 1:200) for 20 minutes.

Ringer's: 128 mM NaCl, 2 mM KCl, 1.8 mM CaCl2, 4 mM MgCl2, 35.5 mM Sucrose, 5 mM Hepes pH 6.9

Fix buffer: 16.7 mM KPO4 pH 6.8, 75 mM KCl, 25 mM NaCl and 3.3 mM MgCl2

PAT: PBS, 1% BSA, 0.1% Triton, 0.05% azide

PBT: PBS, 0.1 BSA, 0.1% tween20

PTW: PBS, 0.1% tween20

1% PBT: 1% Triton x-100 in PBS

0.3% PBTB: 0.3% Triton X-100 and 1% BSA in PBS

PLOS Genetics

## Imaging, processing and quantification of larval, pupal and adult samples

Images were acquired with Leica's TCS-SP5 or Stellaris confocal microscopes, analysed utilizing ImageJ, and processed with Adobe Photoshop and Adobe Illustrator. Z stacks of fixed samples were taken at 0.7 μm intervals using 40x/1.3 and 63x/1.4 NA oil immersion objectives.

Quantification of the number of Vasa+ cells was done as described in [51]. Quantification of the number of Casp3+cells was done manually using the *multi-point* tool in ImageJ.

## Statistical analysis

Experiments involving gonads or ovaries were performed with at least three biological replicates. Samples were collected from at least 5 different larvae, white pupae or adult females grown in equivalent environmental conditions. For each of the quantifications, the arithmetic mean and the standard error of the mean (SEM) of the different experimental settings are shown. Sample sizes correspond to the number of gonads or adult ovaries analysed. Statistically significant differences between control and experimental samples were calculated with a Student's *t*-test. $* = p < 0.05$; $** = p < 0.005$; $*** = p < 0.0005$; $**** = p < 0.0001$. Only significant differences are indicated in the graphs. To test the overlap of different gene sets in Fig 5, we utilized the hypergeometric probability distribution and a permutation test with 10,000 permutations. We assumed a number of detectable D. melanogaster genes of 9,000, presuming that all of the 9,000 genes present in the embryo microarray are also in the gonad microarray and that all are putative targets for the ChIP-seq analyses.

The quantifications of mRNA concentrations using digital PCR were performed with a minimum of two biological replicates.

## Experimental genotypes

Fig 1.

• (F, G) *pnt* enhancer *45E10>tau::GFP: y w/w; P{GMR45E10-GAL4}attP2/ + ; UASp-tau::mGFP6/+*

Fig 2.

• (A, E) control: *w, c587-Gal4/y w;; UASp-tau::mGFP6/+*

• (B, F) *c587>pnt RNAi: w, c587-Gal4/y v;; P{TRiP.JF02227}attP2/+*

• (C, D) control*: c587-Gal4>w RNAi: w, c587-Gal4/y v;; P{TRiP.JF02387}attP2/+*

• *c587>pnt RNAi (HMS01452): w, c587-Gal4/y v;; P{TRiP.HMS01452}attP2/+*

Fig 3.

• (A-L) control: *w;; FRT-82B cu sr pnt^{aga}/TM3, twist-Gal4, UAS-nlsGFP*

• *pnt^{aga}: w;; FRT-82B cu sr pnt^{aga}*

Fig 4.

• (B, D) control: *w;; FRT-82B cu sr pnt^{aga}/TM3, twist-Gal4, UAS-nlsGFP*

*pnt^{aga}: w;; FRT-82B cu sr pnt^{aga}*

• (C) control*: c587-Gal4>w RNAi: w, c587-Gal4/y v;; P{TRiP.JF02387}attP2/+*

• *c587>pnt RNAi (HMS01452): w, c587-Gal4/y v;; P{TRiP.HMS01452}attP2/+*

• *pnt^{aga}/Df: w;; FRT-82B cu sr pnt^{aga}/ Df(3R)ED6105*

- (E, F, G) UASp-pnt$^{intermediate}$; pnt$^{aga}$: UASp-pnt$^{intermediate}$/+; FRT-82B cu sr pnt$^{aga}$

- nos > pnt$^{intermediate}$; pnt$^{aga}$: y w/w; nos-Gal4/UASp-pnt$^{intermediate}$/+; FRT-82B cu sr pnt$^{aga}$

- c587 > pnt$^{intermediate}$; pnt$^{aga}$: w, c587-Gal4/w; UASp-pnt$^{intermediate}$; FRT-82B cu sr pnt$^{aga}$

- UASt-pnt$^{short}$ pnt$^{aga}$: UASt-pnt$^{short}$cu sr pnt$^{aga}$

- UASt-pnt$^{long}$ pnt$^{aga}$: UASt-pnt$^{long}$cu sr pnt$^{aga}$

- (H) control: w, c587-Gal4/+; CyO:GFP/+

- pnt$^{aga}$/Df: w;; FRT-82B cu sr pnt$^{aga}$/ Df(3R)ED6105

- UASt-pnt$^{short}$ pnt$^{aga}$/Df: UASt-pnt$^{short}$, cu sr pnt$^{aga}$/ Df(3R)ED6105

- c587 > pnt$^{intermediate}$; pnt$^{aga}$/ Df(3R)ED6105: w, c587-Gal4/+; UASp-pnt$^{intermediate}$/+; FRT-82B cu sr pnt$^{aga}$/ Df(3R)ED6105

- (I) c587 > pnt RNAi; pnt$^{aga}$: w, c587-Gal4/w; UAS-pnt RNAi$^{KK100473}$/+; FRT-82B cu sr pnt$^{aga}$

Fig 5.

- control: Oregon-R

- pnt$^{aga}$: w;; FRT-82B cu sr pnt$^{aga}$

S1 Fig.

- (A–C) NP5370 > tau::GFP: y w;; P{w[+mW.hs]=GawB} pnt$^{NP5370}$/UASp-tau::mGFP6.

- (D–L) pnt enhancer>tau::GFP: y w/w; P{GMR enhancer-GAL4}attP2/+; UASp-tau::mGFP6/+

- The nine enhancers used are listed in the Fly stocks section above.

S2 Fig.

- (B,C) control: w; +/CyO.

- c587 > pnt RNAi (JF02227): w/y v; tj-Gal4/+; P{TRiP.JF02227}attP2/ +w, c587-Gal4/y v;; P{TRiP. JF02227}attP2/+

- c587 > pnt RNAi (KK100473): w; UAS-pnt RNAi$^{KK100473}$/tj-Gal4

- tj > pnt RNAi (JF02227): w/y v; tj-Gal4/+; P{TRiP.JF02227}attP2/+

- tj > pnt RNAi (KK100473): w; UAS-pnt RNAi$^{KK100473}$/tj-Gal4

- (D) tj > tau::GFP: w/y w; tj-Gal4/+; UASp-tau::mGFP6/+

- (E, F) c587 > tau::GFP: w, c587-Gal4/y w;; UASp-tau::mGFP6/+

- (G, H) nos > tau::GFP: w/y w; nos-Gal4/+; UASp-tau::mGFP6/+

S3 and S4 Figs.

- (A–D) control: y w

- pnt$^{aga}$: w;; FRT-82B cu sr pnt$^{aga}$

S5 Fig.

- (B, C) control: w.

- *pnt^{aga}: w;; FRT-82B cu sr pnt^{aga}*

- (D) *pnt^{Δ88}/Df(3R)6280: y w f/w;; FRT-82B pnt^{Δ88}/Df(3R)Exel6280, P{w[+mC]=XP-U}Exel6280*

- *pnt^{Δ88}/Df(3R)9012: y w f/w;; FRT-82B pnt^{-88}/Df(3R)Exel9012, P{w[+mC]=XP-U}Exel9012*

- *pnt^{Δ88}/Df(3R)ED6105: y w f/w;; FRT-82B pnt^{Δ88}/Df(3R)ED6105, P{w[+mW.Scer\FRT.hs3]=3'.RS5+3.3'}ED6105*

- *pnt^{aga}/ pnt^{Δ88}: w/y w f;; FRT-82B cu sr pnt^{aga}/FRT-82B pnt^{Δ88}*

- *pnt^{aga}/pnt^{1277}: w;; FRT-82B cu sr pnt^{aga}/P{w[+mC]=lacW} pnt^{1277}*

- *pnt^{aga}/Df(3R)9012: w;; FRT-82B cu sr pnt^{aga}/Df(3R)Exel9012, P{w[+mC]=XP-U}Exel9012*

- *pnt^{aga}/Df(3R)6280: w;; FRT-82B cu sr pnt^{aga}/Df(3R)Exel6280, P{w[+mC]=XP-U}Exel6280*

- *pnt^{aga}/Df(3R)ED6105: w;; FRT-82B cu sr pnt^{aga}/Df(3R)ED6105, P{w[+mW.Scer\FRT.hs3]=3'.RS5+3.3'}ED6105*

## Supporting information

**S1 Fig. Expression patterns of several regulatory elements in the *pointed* locus of LL3 gonads and adult ovaries.** In all cases, the regulatory elements drive expression of the Gal4 transcriptional activator. As a reporter, we used the *UASp-tau::GFP* construct. **(A-C)** LL3 gonad, germarium and stage 6 egg chamber of *NP5370>tau::GFP* female larva and adult. They have been stained with anti-Hts (red; to visualise cell outlines and germline spectrosomes and fusomes), anti-Vasa (green; to label the germline), anti-GFP (white; to show Tau::GFP localization) and with a DNA dye (blue; to mark chromatin). The molecular mapping of the NP5370 insertion is represented in the genomic map above the set of panels. **(D-L)** LL3 gonads and germaria of nine different *pnt* enhancer>*tau::GFP* female larvae and adults. They have been stained with anti-Hts (red), anti-Vasa (green) and anti-GFP (white). GSC: germline stem cell; TFC: terminal filament cell; CpC: cap cell; IC: intermingled somatic cell; EC: escort cell; ShC: sheath cell; SwC: Swarm cell. Scale bars: 20 µm. Related to Fig 1.
(TIF)

**S2 Fig. *pointed* RNAi lines and *Gal4* drivers used in this study. (A)** Representation of the exon-intron organization of the *pnt* locus. The three RNAi lines used in this work target similar fragments of the coding region common to all of the transcripts. Numbers of genomic coordinates according to FlyBase (release FB2024_02). **(B)** Quantification of the number of Vasa+ cells in control, *c587>pnt RNAi^{JF02227}* and *c587>pnt RNAi^{KK100473}* LL3 gonads. **(C)** Quantification of the number of Vasa+ cells in control, *tj>pnt RNAi^{JF02227}* and *tj>pnt RNAi^{KK100473}* LL3 gonads. The arithmetic mean and the SEM are shown for each of the genotypes. **(D)** Pattern of expression of the *tj-Gal4* line in control gonads visualised by the distribution of the Tau::GFP reporter (white). **(E, F)** Pattern of expression of the *c587-Gal4* line in control LL3 gonads and adult germaria as shown by the Tau::GFP reporter. The localization of the Hts (red) and Vasa (green) proteins is used to outline cell shapes and to label the germline, respectively. **(G, H)** Pattern of expression of the *nos-Gal4* line in control LL3 gonads and adult germaria as shown by the Tau::GFP reporter. The localization of Hts (red) is used to outline cell shapes. $*=p<0.05$; $**=p<0.005$. Scale bar: 20 µm. Related to Fig 2.
(TIF)

**S3 Fig. *pnt^{aga}* gives rise to agametic testes. (A, B)** dark field images of control (A) and *pnt^{aga}* (B) testes. **(A', B')** Control (A') and *pnt^{aga}* (B') testes stained to visualize F-actin (red) and Vasa (green; to label the germline). *pnt^{aga}* testes are devoid of germline cells. Scale bars: 20 µm. Related to Fig 3. **(C)** Control and **(D)** *pnt^{aga}* LL3 gonads stained with anti-Dcp1 (green; to visualise apoptotic cells), anti-Traffic Jam, anti-Vasa (red; to visualize IC) and with a DNA dye (blue).
(TIF)

**S4 Fig.  *pnt^aga* does not affect PGC numbers in embryonic gonads. (A, B)** Control **(A)** and *pnt^aga* **(B)** embryos at different stages of embryogenesis stained with *D*E-cad (red; to label cell outlines) and Vasa (green; to mark PGCs). **(C)** Quantification of the number of Vasa+ cells in control and *pnt^aga* embryos. **(D)** Quantification of the long, intermediate and short *pnt* mRNA levels in control and *pnt^aga* embryos using droplet-digital PCR. Measurements correspond to two biological and two technical replicates. In spite of no obvious differences in PGC numbers in both genotypes, *pnt* mRNA levels are decreased in *pnt^aga* embryos. Scale bars: 50 μm. Related to Fig 3.
(TIF)

**S5 Fig.  Molecular mapping of *pnt^aga*. (A)** Scheme showing the molecular characterization of *pnt^aga*, *pnt^·88*, *pnt^1277* and the three deficiencies used to map *pnt^aga*. Numbers refer to genomic coordinates according to FlyBase (release FB2024_02). Mapping of *pnt^1277* and the *pnt^·88* deletion according to [11]. **(B)** Sequence of the region around the mapped *pnt^aga* mutation. The "subject" sequence corresponds to the reference sequence in FlyBase (release FB2024_02). The control "query" sequence is that of *y w* flies. The mutant "query" sequence is that of *pnt^aga* flies. **(C)** Chromatogram of the relevant region showing the two-base pair change in *pnt^aga*. **(D)** Table summarising the genetic characterization of different combinations of *pointed* mutants and deficiencies.
(TIF)

**S6 Fig.  Primers used to map *pnt^aga* and associated deficiencies.** The precise mapping of Df(3R)Exel9012, Df(3R)Exel6280 and Df(3R)ED6105 was aided by the transposons used to generate them in the first place. Numbers in parenthesis correspond to the genomic coordinates of the breakpoints. Also shown are the genomic position of the primers used to map *pnt^aga* and the genomic coordinates of the two-base pair substitution found in this mutant.
(TIF)

**S1 Data.  List of differentially expressed genes in control versus *pnt^aga* LL3 gonads.** List of the GO terms identified in the up- or down-regulated genes. Data in this supplementary material refer to Fig 5.
(XLSX)

**S2 Data.  Spreadsheet with numerical data underlying graphs and statistics in Figs 1-4, S2 and S4.**
(XLSX)

## Acknowledgments

We thank the BDSC, the VDRC and the DSHB (University of Iowa) for DNAs, fly stocks and antibodies. The help of Susan Parkhurst and members of her group for discussions and technical support is also acknowledged. We thank members of our group, M.D. Martín-Bermudo, C. Huertas, J. J. Pérez-Moreno, S.C. Herrera, and J. Hombría for comments on the manuscript and helpful discussions. We are grateful to the scientific-technical facilities at the CABD for expert support.

## Author contributions

**Conceptualization:** Acaimo González-Reyes, Alicia E. Rosales-Nieves.

**Data curation:** Alicia E. Rosales-Nieves.

**Formal analysis:** Alicia E. Rosales-Nieves.

**Funding acquisition:** Acaimo González-Reyes.

**Investigation:** Acaimo González-Reyes, Lourdes López-Onieva, Juan Garrido-Maraver, Alicia E. Rosales-Nieves.

**Methodology:** Acaimo González-Reyes, Miriam Marín-Menguiano, Lourdes López-Onieva, Juan Garrido-Maraver, Alicia E. Rosales-Nieves.

**Supervision:** Acaimo González-Reyes, Alicia E. Rosales-Nieves.

**Validation:** Alicia E. Rosales-Nieves.

**Visualization:** Alicia E. Rosales-Nieves.

**Writing – original draft:** Acaimo González-Reyes, Alicia E. Rosales-Nieves.

**Writing – review & editing:** Acaimo González-Reyes, Juan Garrido-Maraver, Alicia E. Rosales-Nieves.

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
