## [Decision Letter · Decision Letter 0]

19 Dec 2024

PGENETICS-D-24-00946

ETS TRANSCRIPTION FACTOR POINTED CONTROLS GERMLINE SURVIVAL IN DROSOPHILA

PLOS Genetics

Dear Dr. Rosales-Nieves,

Thank you for submitting your manuscript to PLOS Genetics. After careful consideration, we feel that it has merit but does not meet PLOS Genetics's publication criteria as it currently stands. All three reviewers have raised significant concerns regarding the genetic characterization of the novel pnt aga mutation. They have suggested several ways to strengthen your conclusions, such as including additional RNAi lines and alleles, testing under different temperatures, and more. Importantly, the reviewers recommend conducting experiments to test the proposed model of regulation between pnt isoforms. We encourage you to submit a substantially revised version of your manuscript that addresses these points, along with the other major concerns highlighted during the review process.

Please submit your revised manuscript within 60 days Feb 17 2025 11:59PM. If you will need more time than this to complete your revisions, please reply to this message or contact the journal office at plosgenetics@plos.org. Please include the following items when submitting your revised manuscript:

We look forward to receiving your revised manuscript.

Kind regards,

Jean-René Huynh

Academic Editor

PLOS Genetics

Aimée Dudley

Editor-in-Chief

PLOS Genetics

Anne Goriely

Editor-in-Chief

PLOS Genetics

**Journal Requirements:**

https://journals.plos.org/plosgenetics/s/submission-guidelines#loc-parts-of-a-submission

4) We have noticed that there is a reference to Suppl. Data 1 in your manuscript. However, there is no corresponding file uploaded to the submission. Please upload it as a separate file with the item type 'Supporting Information'. Please also include it in the legends for your Supporting Information files after the references list.

Potential Copyright Issues:

i) Figures 1C, and 1D. Please confirm whether you drew the images / clip-art within the figure panels by hand. If you did not draw the images, please provide (a) a link to the source of the images or icons and their license / terms of use; or (b) written permission from the copyright holder to publish the images or icons under our CC BY 4.0 license. Alternatively, you may replace the images with open source alternatives. See these open source resources you may use to replace images / clip-art:

6) We note that your Data Availability Statement is currently as follows: "All data and related metadata underlying reported findings will be deposited in appropriate public data repositories." Please confirm at this time whether or not your submission contains all raw data required to replicate the results of your study. Authors must share the “minimal data set” for their submission. PLOS defines the minimal data set to consist of the data required to replicate all study findings reported in the article, as well as related metadata and methods (https://journals.plos.org/plosone/s/data-availability#loc-minimal-data-set-definition).

7) Please amend your detailed Financial Disclosure statement. This is published with the article. It must therefore be completed in full sentences and contain the exact wording you wish to be published.

8) Please ensure that the funders and grant numbers match between the Financial Disclosure field and the Funding Information tab in your submission form. Note that the funders must be provided in the same order in both places as well. Currently, the order of the grants is different in both places.

Please indicate by return email the full and correct funding information for your study and confirm the order in which funding contributions should appear. Please be sure to indicate whether the funders played any role in the study design, data collection and analysis, decision to publish, or preparation of the manuscript.

**Reviewers' comments:**

Reviewer's Responses to Questions

Reviewer #1: The manuscript by Rosales-Nieves et al. highlights a novel role for the ETS transcription factor pointed (pnt) in gametogenesis and reproduction. In Drosophila, the EGFR signaling pathway is known to regulate primordial germ cell (PGC) numbers during female gonad formation, and pnt adds an additional regulatory layer to this process. The authors demonstrate that pnt is expressed in larval and adult gonads and is essential for gametogenesis.

They identified a novel mutant allele of pnt that results in germline loss and the development of agametic gonads. Interestingly, RNAi-mediated downregulation of pnt produces an opposing phenotype, with increased PGC numbers. Sequence analysis was conducted to identify genes affected by this mutation, and the results were compared with previously published data for other pnt mutant alleles. The authors found minimal overlap in differentially expressed genes among these mutants, suggesting that the various pnt isoforms have complex and specific roles in distinct cell types.

This manuscript is well-written, with excellent figures, robust data analysis, and clear conclusions that describe a previously unknown function of the pnt gene. With some revisions and additional experiments, I strongly recommend this paper for publication.

Major Comments:

1. To strengthen the claim that pnt acts in the soma to control germline survival, the use of an additional RNAi line is necessary. The choice to include a second TRIP line for downregulation experiments was commendable, but I recommend using Bloomington stock #35038 from the TRIP collection. This RNAi line, inserted into the VALIUM20 vector, is widely used due to its higher efficiency and lower off-target effects compared to older VDRC Vienna stocks. This line can effectively target both somatic and germline cells. Validation of RNAi efficacy via qPCR is also essential to confirm knockdown efficiency. Furthermore, this new RNAi line could be used for targeted downregulation of pnt specifically in germ cells, addressing whether the observed germline phenotype is consistently reproduced. Since Liu et al., 2010 also used a VALIUM20 RNAi line, testing with the TRIP RNAi line would help verify the phenotype and ensure robustness in the conclusions.

2. To support the hypothesis that pnt isoforms are regulated via an autoregulatory loop, I suggest measuring and comparing the expression levels of pnt isoforms not only in pntaga mutants but also in RNAi mutants that show distinct germline phenotypes. This analysis would provide direct evidence for an autoregulatory mechanism and link the observed germline phenotypes with changes in isoform expression levels.

3. The claim that cell death occurs exclusively in germ cells is not entirely convincing, as intermingled cells also appear to be affected. Can the authors confirm whether intermingled cells are undergoing cell death? This is critical, as intermingled cells are a key source of EGFR signaling, and their loss could disrupt germ cell differentiation and survival. Clarifying this point with appropriate markers or staining would greatly enhance the interpretation of the results.

4. The cause of germ cell loss in the ovary requires further exploration. Given the phenotypic parallels with EGFR signaling and the demonstrated interaction between pnt and this pathway, additional validation would strengthen the argument. Rescue experiments or analyzing reporter expression in the larval ovary could establish a direct link between pnt and EGFR signaling. Additionally, investigating other candidate signaling pathways regulated by pnt would enhance the overall understanding of its role in germ cell survival.

Minor Comments:

5. Line 87: Clarify whether the sentence refers to the yan gene or the Yan protein.

6. Lines 100–102: The statement about terminal filament cells (TFCs) and cap cells (CpCs) in the anterior gonad is inaccurate. The anterior region contains other cell types that differentiate later, not just TFCs. Additionally, CpCs originate from intermingled cells, which are typically not categorized as anterior cells. Please revise accordingly.

7. Figure 1C: While the schematic of the ovary is well-illustrated, there are inaccuracies to address:

i) The anterior region where terminal filament stem cells (TFS) reside is not an empty space; it includes somatic cells like anterior somatic cells and sheath cells. ii) The number of inner germarial (IG) cells depicted does not align with reality; significantly more IG cells are present. iii) Germ cells should not be labeled as primordial germ cells (PGCs) in this stage. Germ cells in contact with the niche are germline stem cells (GSCs), and those further away are differentiated germ cells. Revise this schematic and corresponding text (e.g., line 200) to use "loss of germ cells" as a more accurate term.

8. Line 115: Update the number of ovarioles. Reports indicate there are typically 18–21 ovarioles in D. melanogaster, not 16–18.

9. Line 152: Explain why UASp-Tau::GFP was chosen instead of the standard pUASp-GFP.

10. Line 155: Provide more specific details on pnt reporter's expression pattern. In larval ovaries, the pattern is restricted to a subset of TFCs near GSCs and CpCs. In adult ovaries, expression is limited to the anterior TFC and the last TFC in contact with CpCs. Specifying these differences is important, since TFCs exhibit distinct signaling pathway expression patterns.

11. Lines 174–180: These sentences are unclear and need rephrasing for better readability and understanding.

12. Cell Proliferation: Confirm cell proliferation using specific markers, rather than relying solely on cell counts. This would add robustness to the data.

13. Figures 3H and 4A: While germ cell-positive markers are absent in the germarium, some cells with large nuclei appear in the escort cell region. Could the authors comment on the identity of these cells?

14. Line 196: Clarify what n refers to in experiments—does it indicate the number of flies or ovaries analyzed? This specification is critical for understanding experimental design.

Reviewer #2: The work of A. Rosales-Nieves et al. focuses on a genetic study of Pointed transcription factor role in the development of gonads in Drosophila. The research highlights two main observations. On the one hand, the expression levels of pointed are critical, as a significant loss leads to an increase in primordial germ cells, while a more limited loss results in their disappearance. On the other hand, the results suggest that pointed isoforms regulate each other. This work needs to be further consolidated before considering its publication. Moreover, its relevance for a large audience is not entirely obvious to me.

1. The increase in the number of PGCs is observed in two different RNAi constructs, but these transgenes target highly overlapping sequences, which does not rule out the possibility of a common off-target effect. Authors should find a way to exclude it. Moreover, a control RNAi (white, yellow, luc…) would be better rather than just comparison to Gal4 for the negative control.

2. Conversely, the aga mutant results in the absence of PGCs. However, according to Figure S5, it does not show a phenotype with the null mutant delta88, while it fails to complement larger deletions. The simplest interpretation would be that this allele (or another mutation on the same chromosome) affects another gene covered by the deficiency. Moreover, the authors provide no explanation regarding the origin of this mutation. If it originates from a genetic screen where flies were mutagenized, the existence of other mutations is likely. I am fully aware that the authors achieved rescue of this allele with transgenes, but the overall picture lacks coherence. Could it be that the phenotype is related to the concomitant loss of pointed and another gene?

3. The authors' results suggest that the short isoform might activate the transcription of the long isoform. This hypothesis is easy to test: In their rescue experiment, does the expression of the short isoform increase that of the long one?

4. The authors compare their differential expression results with those of other studies and deduce common targets. Does this correspond to a statistically significant enrichment? In other words, if random gene lists of the same size were intersected, what would be the expected number of common genes?

Minor points :

- Scale bars are missing on many pictures.

- Figure S1 shows that the locus seems to contain many enhancers leading to rather similar expression pattern but these results are poorly described. Is there an explanation for this? Could it be related to the fact that the isoforms seem to control each other?

5. Is there any evidence for a similar short isoform in mammals?

Reviewer #3: In this article, Alicia Rosales-Nieves and colleagues investigate the function of the ETS transcription factor Pointed during female gonad development in Drosophila.

Pointed belongs to the ETS domain family, a large group of transcription factors. Most ETS proteins are nuclear targets of Ras-MAP kinase signalling pathways, in particular upon activation of the EGF tyrosine kinase receptor.

The authors investigated the involvement of pointed (pnt) in the formation of the female gonads, following previous work indicating that silencing of somatic EGFR signaling causes overproliferation of primordial germ cells (PGCs).

They first used digital droplet PCR to precisely document the expression profile of the three pnt isoforms, identified as long, intermediate and short. In the L3 female larval gonad, only pnt short and long are expressed, while only pnt long is expressed in the ovarian tissues, including the germaria and early egg chambers. To further analyse pnt expression in female gonads, the authors used several enhancer-Gal4 lines and found that pnt expression is restricted to somatic cells of L3 female gonads and the ovarian GSC niche.

To assess the requirement for pnt during early oogenesis, they combine several RNAi lines whose expression is restricted to the somatic cells of the gonads and the germarium. In female larval gonads, somatic reduction of pnt expression leads to an increase in PGCS. In the adult, the authors report that somatic reduction of pnt expression in the ovary leads to larger germaria with more germ cells. They also show that reducing pnt in somatic cells impairs their ability to correctly encapsulate the GC cyst.

The authors have isolated and characterised a new pnt mutation which, according to their results, specifically affects the development of PGCs from the L3 larval stage, with a significant decrease in the number of PGcs due to apoptosis in PGCs.

They further characterised the pnt aga mutation and found that it corresponds to a two-nucleotide change that affects pointed transcription.

Finally, they performed a comparative transcriptomic analysis between wild-type and pnt aa LL3 gonads and embryos.

Overall, this is a very comprehensive, multi-scale study with sophisticated and carefully chosen tools. The experiments are carefully made, and most of the data are presented in a clean nicely organized manner, well controlled and carefully quantified.

Main point

This work provides new insights into the role of the EGF pathway and the ETS transcription factor, pointed in the control of PGC homeostasis in the female gonad.

However, the effect of the pnt aga mutation which is central in this manuscript is difficult for me to understand. On the one hand, molecular analysis seems to indicate a reduction in transcript levels for pnt long and pnt short.

On the other hand, the nature of the defects observed with pnt aga are opposite to those observed with pnt RNAi and are consistent with observations when the EGF pathway is inactivated: disappearance of CPGs with pnt aga when reduction of pnt by RNAi induces an excess of CPGs. Pnt aga seems to act as a gain of function. This is also consistent with the fact that pnt RNA i rescues the defects observed with pnt aga. Similarly, transcriptomic analysis of Pntaga embryos shows that 3 genes (hibris, cg1773 and 3 3- Hydroxymethyl-3-methylglutaryl-CoA lyase ) are overexpressed, whereas these genes are underexpressed in pnt mutant embryos. This possibility is never mentioned by the authors.

The authors suggest that the difference in defects observed is related to the level of pnt transcripts. A large decrease, as with RNAi, would lead to an overproliferation of PGCs, whereas a decrease of 50% would have the opposite effect, the loss of PGCs.

To test this possibility, the authors could reduce the efficiency of RNAi by lowering the induction temperature of gal4 or by using another gal4 line that is less expressed in somatic gonadal tissues. This can be monitored by ddPCR, as the authors have done for pnt aga.

By focusing on gonad development at stage L3, the reduced efficiency in reducing the levels of short pnt and long pnt transcripts should lead to the loss of PGCs, as in the case of pnt aga.

Minor points.

In the Figure 2, the increase of germline cell upon pnt rnai in somatic adult female gonads is difficult to distinguish, so a close up may help.

In the figure 3, with the panel I, the number of germ cells in embryos is very small this is probably due to the scale of the representation. It would be possible to modify the representation of these values with a different scale and insert them to complement figure sup S4.

**Have all data underlying the figures and results presented in the manuscript been provided?**

Reviewer #1: Yes

Reviewer #2: Yes

Reviewer #3: Yes

PLOS authors have the option to publish the peer review history of their article (what does this mean? ). If published, this will include your full peer review and any attached files.

**Do you want your identity to be public for this peer review?** For information about this choice, including consent withdrawal, please see our Privacy Policy .

Reviewer #1: No

Reviewer #2: No

Reviewer #3: No

**Figure resubmission:**
---

## [Decision Letter · Decision Letter 1]

15 Jul 2025

Dear Dr Rosales-Nieves,

We are pleased to inform you that your manuscript entitled "ETS TRANSCRIPTION FACTOR POINTED CONTROLS GERMLINE SURVIVAL IN DROSOPHILA" has been editorially accepted for publication in PLOS Genetics. Congratulations!

Yours sincerely,

Jean-René Huynh

Academic Editor

PLOS Genetics

Paula Cohen

Section Editor

PLOS Genetics

Aimée Dudley

Editor-in-Chief

PLOS Genetics

Anne Goriely

Editor-in-Chief

PLOS Genetics

Comments from the reviewers (if applicable):

Reviewer's Responses to Questions

**Comments to the Authors:**

Reviewer #1: The authors have fully addressed all my previous concerns and performed additional experiments that convincingly strengthen their conclusions. The study is well-written, with beautiful figures, robust analyses, good genetics, and clear conclusions describing a previously unknown role of pnt. I strongly recommend this paper for publication.

Reviewer #2: The authors answered all my questions appropriately. The current version of the article has been significantly improved.

Reviewer #3: In this revised manuscript, Alicia Rosales-Nieves and her colleagues have significantly improved their manuscript. They have performed new experiments to address several points raised by the reviewers. The authors have extensively modified the figures by integrating these new results and improving the presentation of the results to enhance their significance.

In general, the authors have addressed and responded satisfactorily to the various points raised, clearly explaining where points could not be addressed due to experimental limitations. They have also provided detailed information on the comments and questions raised by the reviewers.

The manuscript has improved since the last submission and I strongly recommend its publication in PLOS Genetics.

**Have all data underlying the figures and results presented in the manuscript been provided?**

Reviewer #1: Yes

Reviewer #2: Yes

Reviewer #3: Yes

PLOS authors have the option to publish the peer review history of their article (what does this mean? ). If published, this will include your full peer review and any attached files.

**Do you want your identity to be public for this peer review?** For information about this choice, including consent withdrawal, please see our Privacy Policy .

Reviewer #1: No

Reviewer #2: No

Reviewer #3: No

**Data Deposition**

http://datadryad.org/submit?journalID=pgenetics&manu=PGENETICS-D-24-00946R1

**Press Queries**

---

## [Editor Report · Acceptance letter]

PGENETICS-D-24-00946R1

ETS TRANSCRIPTION FACTOR POINTED CONTROLS GERMLINE SURVIVAL IN DROSOPHILA

Dear Dr González-Reyes,

We are pleased to inform you that your manuscript entitled " 

ETS TRANSCRIPTION FACTOR POINTED CONTROLS GERMLINE SURVIVAL IN DROSOPHILA" has been formally accepted for publication in PLOS Genetics! Your manuscript is now with our production department and you will be notified of the publication date in due course.

With kind regards,

Anita Estes

PLOS Genetics

On behalf of:
